# QVLA: NOT ALL CHANNELS ARE EQUAL IN VISION-LANGUAGE-ACTION MODEL'S QUANTIZATION

**Yuhao Xu**[1,3,4*]   **Yantai Yang**[1,2*]   **Zhenyang Fan**[1]   **Yufan Liu**[3,4†]
**Yuming Li**[5]   **Bing Li**[3†]   **Zhipeng Zhang**[1†]

[1]AutoLab, School of Artificial Intelligence, Shanghai Jiao Tong University [2]Anyverse Dynamics
[3]State Key Laboratory of Multimodal Artificial Intelligence Systems,
Institute of Automation, Chinese Academy of Sciences
[4]School of Artificial Intelligence, University of Chinese Academy of Sciences
[5]Terminal Technology Department, Alipay, Ant Group
 **Code:** https://github.com/AutoLab-SAI-SJTU/QVLA

## ABSTRACT

The advent of Vision-Language-Action (VLA) models represents a significant leap for embodied intelligence, yet their immense computational demands critically hinder deployment on resource-constrained robotic platforms. Intuitively, low-bit quantization is a prevalent and preferred technique for large-scale model compression. However, we find that a systematic analysis of VLA model's quantization is fundamentally lacking. We argue that naively applying *uniform-bit quantization* from Large Language Models (LLMs) to robotics is flawed, as these methods prioritize passive data fidelity while ignoring how minor action deviations compound into catastrophic task failures. To bridge this gap, we introduce QVLA, the first action-centric quantization framework specifically designed for embodied control. In a sharp departure from the rigid, *uniform-bit* quantization of LLM-based methods, QVLA introduces a highly granular, *channel-wise bit allocation* strategy. Its core mechanism is to directly measure the final action-space sensitivity when quantizing each individual channel to various bit-widths. This process yields a precise, per-channel importance metric that guides a global optimization, which elegantly *unifies quantization and pruning (0-bit) into a single, cohesive framework*. Extensive evaluations on different baselines demonstrate the superiority of our approach. In the LIBERO, the quantization version of OpenVLA-OFT with our method requires only **29.2%** of the original model's VRAM while maintaining **98.9%** of its original performance and achieving a 1.49× speedup. This translates to a **22.6%** performance improvement over the LLM-derived method SmoothQuant. Our work establishes a new, principled foundation for compressing VLA models in robotics, paving the way for deploying powerful, large-scale models on real-world hardware.

## 1 INTRODUCTION

The rapid evolution of foundation models (Touvron et al., 2023a;b) has significantly advanced embodied intelligence, empowering Vision-Language-Action (VLA) models like OpenVLA (Kim et al., 2024) to synthesize complex, executable actions from visual inputs and linguistic directives. These models enhance cross-task generalization and semantic reasoning, elevating the efficacy of robotic manipulation. However, their immense computational and memory demands, which often exceed 14 GB in standard half-precision for a 7B model, present a critical barrier to deployment. For instance, running such a model on a widely-used robotic platform like the NVIDIA Jetson AGX Orin can result in inference latencies of several hundred milliseconds per action, far too slow for the fluid, real-time control required in dynamic environments. This performance gap necessitates aggressive compression methodologies, such as pruning (Liu et al., 2022b; Ruan et al., 2020; 2021), knowledge distillation (Liu et al., 2019; 2022a) and quantization (Xiao et al., 2022; Lin et al., 2023), to achieve

---

*Equal contribution. †Corresponding author.   Work performed during a remote internship at SJTU.

practical inference speeds while sustaining precise control (Yang et al., 2025; Wen et al., 2024; Song et al., 2025b;a). Surprisingly, while low-bit quantization is a well-established technique extensively studied in Large Language Models (LLMs), we find there has been no systematic analysis of its unique impacts and trade-offs when applied specifically to VLA methods.

This gap is not merely an academic oversight but a critical barrier, as naively applying existing quantization techniques ignores the fundamental distinctions between VLA models and their LLMs or Multimodal Large Language Models (MLLMs) counterparts. Indeed, as aforementioned, the development of quantization techniques has been predominantly driven by the requirements of LLMs (Frantar et al., 2022; Li et al., 2021; Nagel et al., 2020) and general-purpose MLLMs (Wang et al., 2024). These approaches are optimized to preserve text perplexity or visual feature fidelity, often using proxy loss functions. In stark contrast, the output of a VLA model is not passive text or a label, but a sequence of continuous action values that directly interface with the physical world. In this closed-loop setting, even subtle quantization-induced errors in action outputs, that may be imperceptible in standard benchmarks, can be amplified by physical dynamics and contact forces. Over a long-horizon task, these errors accumulate autoregressively, leading to catastrophic failures such as unstable grasps or significant trajectory deviations (as illustrated in Fig. 3). Consequently, directly porting quantization frameworks designed for passive data processing is fundamentally ill-suited for the demands of active, embodied control, as it often undermines the requisite stability and precision. This naturally raises the question that *How should one design a quantization method specifically tailored to the unique demands of VLA models?*

Before answering this question, we first revisit the predominant paradigms in model quantization. Recent scholarly efforts, exemplified by representative works such as SmoothQuant (Xiao et al., 2022), have primarily focused on outlier management. These methods employ techniques like rotations, permutations, or saliency-based protections to mitigate the dominance of extreme values in quantization step sizes, thereby enhancing low-bit performance. However, this outlier-centric paradigm proves insufficient for VLA models, where cross-modal alignment and action decoding interfaces (*e.g.*, projectors and action heads) exhibit acute sensitivity to perturbations, while long-horizon tasks amplify even minor initial errors (see analysis in Sec. 3.2 and again Fig. 3). Concurrently, in industrial practice, module-level mixed precision has emerged as a compromise, such as quantizing vision encoders to 4-bit while preserving language backbones at 8-bit. Yet, this coarse-grained approach lacks the necessary precision. Our analysis, as detailed in Sec. 3.2, reveals significant intra-layer channel heterogeneity. Specifically, individual channels contribute variably to the final VLA outputs, which is a critical distinction that conventional methodologies fail to address.

Building upon this foundational analysis, we present QVLA, an advanced, action-centric framework for channel-wise quantization. To our knowledge, this marks the first systematic exploration of quantization specifically tailored for VLA architectures. Our method directly anchors the quantization objective within the action space, rather than the internal representation. Moreover, our approach *naturally unifies quantization and pruning (0-bit) into a single process*, enabling fine-grained, per-channel bit allocation. These advancements are achieved in two key steps: (1) **Action-space sensitivity estimation** that measures how much quantizing each channel affects the final action output with the proposed sensitivity metrics. For efficiency, we use a fast first-order proxy based on Taylor-series approximation to identify the most sensitive channels. (2) **Optimal bit allocation** that assigns the final bit-widths in $\{0, 2, 4, 8, 16\}$ to each channel with the proposed global greedy demotion algorithm, starting from full precision and progressively lowering the bit-width of the least sensitive ones until the budget is met. Experiments on OpenVLA (Kim et al., 2024) and OpenVLA-OFT (Kim et al.) baselines validate the efficacy of QVLA. In the LIBERO environment, our method on the OpenVLA-OFT requires only **29.2%** of the original model's VRAM while maintaining **98.9%** of its original performance and achieving a $1.49\times$ speedup.

Our contributions include: ♠ We conduct the first systematic analysis of quantization challenges unique to VLA models. Our findings reveal why existing paradigms fail and establish that aligning quantization with the action space is a foundational principle for effective compression in embodied AI. ♠ We propose QVLA, a novel channel-wise framework that uniquely uses action-space sensitivity to guide bit allocation, cohesively unifying weight quantization and pruning (0-bit). ♠ Through extensive evaluations on OpenVLA and OpenVLA-OFT baselines, we demonstrate that QVLA significantly outperforms methods adapted from LLMs and MLLMs. At equivalent average bit-widths, our framework achieves substantially lower action errors and higher task success rates, validating its efficacy for robotics in resource-constrained environments.

## 2 RELATED WORK

**Vision-Language-Action models.** Vision-Language-Action (VLA) models represent a dominant paradigm for generalist robotic control, learning direct policies that map high-dimensional visual observations and language instructions to low-level motor commands. Methodologies for VLA development have largely bifurcated based on the action decoding strategy. The first, exemplified by models like RT-2 (Zitkovich et al., 2023), OpenVLA (Kim et al., 2024), and UniVLA (Bu et al., 2025), discretizes the continuous action space, casting control as a sequence-to-sequence problem. Conversely, a second line of work prioritizes temporal fidelity and high-frequency control by modeling actions in the continuous domain. This is typically achieved with powerful generative decoders, such as the diffusion policies in Octo (Octo Model Team et al., 2024) and RDT-1B (Liu et al., 2024b) or the flow-matching network in $\pi_0$ (Black et al., 2024). Despite their distinct advantages, both approaches are encumbered by a substantial computational footprint (Shukor et al., 2025; Yang et al., 2025), rendering their large-scale architectures prohibitive for real-time execution on resource-constrained robotics hardware. While architectural compression, TinyVLA (Wen et al., 2024), offers a partial solution, a more fundamental optimization strategy from the broader deep learning field, *i.e.,* low-bit quantization, remains conspicuously under-explored within embodied AI. This work aims to fill this critical gap, positing that quantization is not merely an incremental optimization but a foundational component required to unlock the practical deployment of generalist VLA models.

**Quantization methods.** Model quantization is a cornerstone technique for efficient deep learning deployment, reducing memory footprint and computational latency by representing weights and activations with low-bit integers. The central challenge lies in minimizing the ensuing accuracy degradation. Two primary methodologies dominate the field, *i.e.,* Post-Training Quantization (PTQ) (Frantar et al., 2022; Li et al., 2021; Nagel et al., 2020) and Quantization-Aware Training (QAT) (Jacob et al., 2018; Esser et al., 2020). PTQ offers a low-cost solution by quantizing a pretrained model with a small calibration set and no retraining. In contrast, QAT simulates the effects of quantization during training, allowing the model to adapt its parameters to mitigate precision loss. Recent advances have focused on mitigating the challenges of quantizing large models, particularly the presence of outliers and activation-induced quantization difficulties. Methods like SmoothQuant (Xiao et al., 2022) re-scale weights and activations to create a more favorable quantization landscape, while AWQ (Lin et al., 2023) preserves salient weights and OmniQuant (Shao et al., 2024) smooths weight and activation outliers. To handle pernicious outliers in weights and activations, another research thrust employs learned rotations or permutations to redistribute value distributions, exemplified by methods like QuaRot (Ashkboos et al., 2024) and SpinQuant (Liu et al., 2024c), enabling robust 4-bit quantization. However, a common limitation across these methods is the assumption of uniform precision, where a single bit-width is applied globally or, at best, per layer (*e.g.,* HAWQ (Dong et al., 2019; 2020)). This coarse-grained approach is fundamentally misaligned with the requirements of embodied policies. VLA models exhibit heterogeneous sensitivity across their architecture. For instance, subtle shifts in action-generation layers can lead to catastrophic failures in control, a fragility not typically observed in standard perception or language tasks. Motivated by this critical shortcoming, we propose a fine-grained, action-guided, channel-wise quantization framework.

## 3 METHOD

### 3.1 PRELIMINARIES

**Vision-Language-Action (VLA) models.** A VLA model governs the behavior of an embodied agent by defining a policy, denoted as $\Pi$, that maps high-dimensional sensory inputs and a language directive to a sequence of actions. The model operates within a sequential decision-making framework. Formally, at each discrete timestep $t$, the agent perceives its environment through visual observations $\mathcal{V}_t$ (*e.g.,* RGB images) and receives a time-invariant language instruction (prompt) $p$. The policy's role is to predict an action sequence $\mathcal{A}_t$ (*e.g.,* end-effector poses or joint velocities),

$$\Pi_\theta(\mathcal{A}_t | \mathcal{V}_t, p). \tag{1}$$

The parameter set $\theta$ typically comprises: ♠ vision encoders $\theta_{vis}$ that processes the high-dimensional visual inputs $\mathcal{V}_t$ into compact feature representations; ♠ a projection layer $\theta_{proj}$ that maps these visual features into the multimodal embedding space shared with the language modality; ♠ a Large

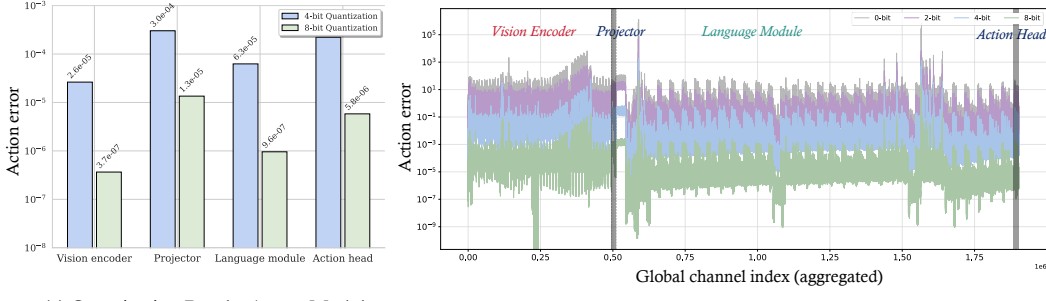

(a) Quantization Results Across Modules     (b) Quantization Results Across Channels

Figure 1: Quantitative analysis of quantization sensitivity in VLA models. **(a)** Per-module analysis shows that the projector and action head are significantly more sensitive to quantization. **(b)** Per-channel analysis demonstrates highly heterogeneous sensitivity within modules. These findings collectively motivate our adaptive precision, channel-level quantization approach.

Language Model (LLM) decoder $\theta_{llm}$, which serves as the core reasoning engine, contextualizing the visual percepts with the task prompt $p$; and ♠ an action decoder $\theta_{act}$ that translates the final latent representation from the LLM into an executable action sequence $\mathcal{A}_t$. The learnable parameters $\theta = \{\theta_{vis}, \theta_{proj}, \theta_{llm}, \theta_{act}\}$ of this policy are the subject of our investigation into model compression via quantization. A central challenge, which the VLA quantization method is designed to address, is the pronounced sensitivity of the policy's action output, $\mathcal{A}_t$, to perturbations in its parameters $\theta$.

**Model quantization.** Quantization is the process of mapping the continuous, full-precision parameter set $\theta$ to a discrete, low-precision set $\hat{\theta}$ by a quantization function $\hat{\theta} = Q(\theta)$. The fundamental objective of quantization is to find an optimal mapping $Q^*$ that minimizes the performance degradation caused by the reduced precision. In our VLA context, this is equivalent to minimizing the divergence between the action distributions of the original policy $\Pi_\theta$ and the quantized policy $\Pi_{\hat{\theta}}$,

$$Q^* = \arg\min_{Q} \quad \mathbb{E}_{(\mathcal{V}_t, p, \mathcal{H}_t) \sim \mathcal{D}} \left[ D_{KL} \left( \Pi_\theta(a_t | \mathcal{V}_t, p, \mathcal{H}_t) \,\|\, \Pi_{Q(\theta)}(\mathcal{A}_t | \mathcal{V}_t, p, \mathcal{H}_t) \right) \right], \quad (2)$$

where $\mathcal{D}$ is the data distribution and $D_{KL}$ denotes the Kullback-Leibler divergence. Conventional quantization approaches usually apply a *uniform bit-width* across all parameters. In such a scheme, for a given weight tensor $\mathbf{W}$ from a linear transformation $\mathbf{Y} = \mathbf{XW} + \mathbf{b}$, the quantization process involves computing a scaling factor $\alpha_W$ for the entire tensor and then mapping the full-precision values to a single $k_w$-bit integers, using nearest rounding and a clamping function,

$$\mathbf{W}_q = \text{clamp} \left( \left\lfloor \frac{\mathbf{W}}{\alpha_W} \right\rceil, -2^{k_w-1}, 2^{k_w-1} - 1 \right). \quad (3)$$

The de-quantized weights $\hat{\mathbf{W}} = \mathbf{W}_q \cdot \alpha_W$ are then used to approximate the original computation, *i.e.*, $\mathbf{Y} \approx \hat{\mathbf{X}}\hat{\mathbf{W}} + \mathbf{b}$, where $\hat{\mathbf{X}}$ is the similarly quantized activation tensor. While simple to implement, this uniform strategy does not account for the heterogeneous sensitivity of different parameters to quantization noise. Therefore, our work is to find a more effective approximation of $Q^*$ beyond the uniform bit-width that preserves the crucial behavioral characteristics of the agent.

## 3.2 SENSITIVITY ANALYSIS

**Not all modules are equal.** As previously delineated, VLA models possess a modular architecture. A naive application of uniform bit-width quantization across these diverse modules yields suboptimal results. Our empirical investigation, summarized in Fig. 1(a), systematically illustrates this disparity by isolating the quantization of each module and measuring the resultant impact on task performance. More concretely, the vision encoder ($\theta_v$) demonstrates considerable robustness than other components. This resilience likely stems from the high-dimensional and often redundant nature of visual input, where feature representations are more tolerant to perturbations. In contrast, the language module ($\theta_l$) proves to be more vulnerable, as has also been noted in prior work (Jiang et al., 2025; Park et al., 2024). Quantizing this component to the same bit-width leads to a more pronounced performance degradation. Most critically, the cross-modal interfaces, *i.e.,* the projector ($\theta_p$) and the action head ($\theta_a$), exhibit the most acute sensitivity. Aggressive quantization in these modules precipitates a severe, often catastrophic, decline in performance. This is because these

components serve as the final nexus for translating multimodal understanding into physical action. Any perturbations introduced here propagate directly and without mitigation to the output action distribution, leading to significant and often erroneous deviations in the agent's behavior. These findings underscore that an adaptive quantization strategy is not merely beneficial but essential for preserving the functional integrity of VLA models at low bit-widths.

**Not all channels are equal.** Motivated by this module-level disparity, we extend our analysis to a finer granularity, revealing a pronounced heterogeneity even among channels within the same layer. Here, a "channel" refers to an output channel in a convolutional layer or a row in the weight matrix of a linear layer. As illustrated in Fig. 1(b), the impact of quantization on action errors is not uniformly distributed across channels. This observation highlights the inherent limitations of uniform bit allocation schemes (whether global or per-layer) and strongly motivates a more nuanced, per-channel mixed-precision strategy, which could include pruning (*i.e.*, 0-bit quantization) for the least sensitive channels to optimize the computational budget.

To effectively identify these salient channels, we propose a novel sensitivity metric grounded directly in the action space rather than the intermediate feature space. More specifically, for a given layer $l$ and a specific channel $c$ within it, we isolate its impact by quantizing only that channel to a bit-width $b \in \{0, 2, 4, 8, 16\}$, while all other parameters remain in full precision. We then define the single-step action sensitivity as the expected squared L2 norm of the resulting action deviation,

$$s_{l,c}^{(b)} = \mathbb{E}_{x \sim \mathcal{D}} \left[ \left\| \tilde{\mathcal{A}}_{l,c}^{(b)}(\mathcal{V}, l) - \mathcal{A}^*(\mathcal{V}, l) \right\|_2^2 \right], \tag{4}$$

where $\tilde{\mathcal{A}}_{l,c}^{(b)}(x)$ is the perturbed action generated with the quantized channel, and $\mathcal{A}^*(x)$ is the reference action from the full-precision model. However, single-step error metrics like $s_{l,c}^{(b)}$ may not fully capture the error accumulation that occurs in long-horizon, autoregressive tasks. A small, initial error can compound over a sequence of actions. To account for this, we introduce a cumulative sensitivity metric, which measures the total deviation over an entire episode:

$$S_{l,c}^{(b)} = \mathbb{E} \left[ \sum_{t=1}^{T} \left\| \tilde{\mathcal{A}}_{l,c}^{(b)}(\mathcal{V}_t, l) - \mathcal{A}^*(\mathcal{V}_t, l) \right\|_2 \right]. \tag{5}$$

This cumulative metric, $S_{l,c}^{(b)}$, naturally exhibits a stronger correlation with ultimate task success.

## 3.3 THE PROPOSED QVLA

Building upon the insights from our multi-granularity sensitivity analysis, we introduce QVLA, a quantization framework specifically designed to address the acute sensitivity of VLA outputs and ensure robust action generation. Departing from conventional LLMs and MLLMs quantization approaches that focus on reconstructing internal feature representations, the proposed QVLA framework emphasizes the preservation of action fidelity, aligning quantization directly with the functional objectives of VLA models. To facilitate a hardware-friendly implementation, we adopt *uniform-bit activations* across the model, a pragmatic choice that avoids the runtime branching and kernel fragmentation that can degrade computational performance. In contrast, weights receive a more fine-grained, *per-output-channel integer* quantization. To enable this, all operators are first standardized as linear maps of the form $\mathbf{Y} = \mathbf{XW} + \mathbf{b}$, where convolutions are treated as their equivalent linear operators. Each output channel can then be assigned a unique bit-width from $\{0, 2, 4, 8, 16\}$, a formulation that elegantly unifies quantization with structural pruning by treating a 0-bit assignment as a channel to be pruned. Finally, performance is evaluated directly within the action space to measure true task-level impact. More specifically, single-step accuracy is quantified using Action-MSE under a teacher-forcing paradigm, while temporal robustness is assessed through short-horizon rollouts by measuring cumulative action and end-effector deviations alongside final task success rates. Fig. 2 shows our overall framework.

### 3.3.1 ESTIMATION OF ACTION-SPACE SENSITIVITY

Our bit allocation strategy is guided by the action-space sensitivity metric defined in Eq. 4 and 5. For each channel $(l, c)$, we compute a set of sensitivity scores, $s_{l,c}^{(b)}$, by evaluating its impact at

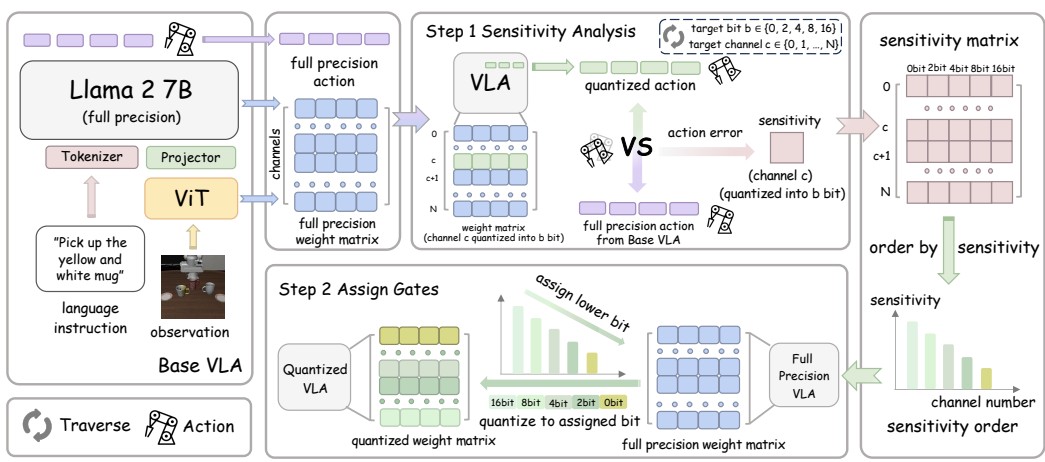

Figure 2: The pipeline of our **QVLA** framework consists of two steps: (i) In Step 1, we conduct a fine-grained action sensitivity analysis by systematically measuring and ranking the error induced by quantizing each channel to various bit-widths. (ii) In Step 2, an optimal bit-width is assigned to each channel using a greedy demotion algorithm, which iteratively prunes or lowers the precision of the least sensitive channels until the target bit budget is met.

different bit-widths $b \in \{0, 2, 4, 8, 16\}$ on a calibration set $\mathcal{D}$. A crucial property of these scores is their inherent comparability across all network components (modules, layers, and channels), which enables them to serve as the primary signal for our global bit allocation algorithm. To ensure this single-step metric is also a valid proxy for performance in dynamic control tasks, we introduce a cumulative variant, $S_{l,c}^{(b)}$, designed to capture short-horizon error accumulation. In practice, we find that the rankings of channel sensitivities produced by $s_{l,c}^{(b)}$ and $S_{l,c}^{(b)}$ are highly consistent. This crucial finding allows us to leverage the computationally cheaper single-step metric $s_{l,c}^{(b)}$ for guiding the bit allocation process, while using the more comprehensive cumulative metric $S_{l,c}^{(b)}$ to validate that our approach successfully extrapolates to long-horizon performance.

While the sensitivity metric $s_{l,c}^{(b)}$ provides a robust signal for bit allocation, exhaustively computing it for every channel and bit-width is computationally prohibitive. To overcome this challenge, we introduce an efficient two-stage strategy that combines a rapid, approximate screening with a targeted, precise evaluation. First, we derive a first-order approximation to serve as a proxy for sensitivity. The core idea is to model the local relationship between a channel's output, $\mathbf{X}_{l,c}$, and the final action, $\mathcal{A}$, using a linear approximation based on the Taylor expansion. A small perturbation $\Delta\mathbf{X}_{l,c}$ at the channel output will induce a deviation in the action, $\Delta\mathcal{A}$, which can be approximated as $\Delta\mathcal{A} \approx J_{\mathcal{A},\mathbf{X}_{l,c}}\Delta\mathbf{X}_{l,c}$. To quantify the magnitude of this effect, we consider the vector norms,

$$\|\Delta\mathcal{A}\| \approx \|J_{\mathcal{A},\mathbf{X}_{l,c}}\| \cdot \|\Delta\mathbf{X}_{l,c}\|, \tag{6}$$

where $J_{\mathcal{A},\mathbf{X}_{zu_{l,c}}}$ is the Jacobian of the action with respect to the channel output. In this formulation, the matrix norm $\|J_{\mathcal{A},\mathbf{X}_{l,c}}\|$ serves as a local sensitivity gain, a scalar value that quantifies how much a perturbation's magnitude is amplified as it propagates to the action space. The perturbation induced by quantization is modeled as the quantization error, $\Delta\mathbf{X}_{l,c} \approx (\mathrm{Q}(\mathbf{W}_l) - \mathbf{W}_l)\mathbf{X}_l$. This allows us to compute a rapid importance score, *i.e.,* the product of the Jacobian gain and the estimated quantization noise, to create a global ranking of all channels. Then, based on this ranking, we perform a limited number of full forward passes, selectively targeting the most important channels to precisely calibrate their true sensitivity scores. This hybrid approach allows us to focus computational effort on preserving sensitive interfaces, such as the projector and action head, while enabling aggressive compression of less critical channels.

### 3.3.2 OPTIMAL BIT ALLOCATION UNDER CONSTRAINED BUDGET

With the sensitivity scores for each potential bit-width established, we can now formulate the overall bit allocation task as a constrained optimization problem. Specifically, we aim to assign a bit-width

Table 1: **Performances under various weight-activation quantization settings**. W4A4/W8A8 refers to the quantization of weights (W) and activations (A) to 4 and 8 bits, respectively. Note that since our method assigns bits adaptively on a per-channel basis, we report the average bit-width.

| Model | Setting | Method | Spatial | Object | Goal | Long | Avg ↑ | Δ | Mem. (GB) ↓ | Speedup ↑ |
|---|---|---|---|---|---|---|---|---|---|---|
| OpenVLA | FP Model | - | 84.7% | 88.4% | 79.2% | 53.7% | 76.5% | - | 15.2 | 1× |
| | W8A8 | SmoothQuant | 84.2% | 87.8% | 77.8% | 53.2% | 75.8% | -0.7% | 7.4 | 1.40× |
| | | OmniQuant | 82.6% | 86.2% | 74.8% | 51.7% | 73.8% | -2.7% | 7.8 | 1.26× |
| | | QVLA | 85.2% | 88.0% | 77.6% | 54.2% | 76.3% | -0.2% | 7.1 | 1.42× |
| | W4A4 | SmoothQuant | 69.2% | 73.2% | 69.6% | 40.9% | 63.2% | -13.3% | 4.7 | 1.52× |
| | | OmniQuant | 82.2% | 85.4% | 75.4% | 50.3% | 73.3% | -3.2% | 5.4 | 1.43× |
| | | QVLA | 84.4% | 87.6% | 78.8% | 53.0% | 76.0% | -0.5% | 4.3 | 1.47× |
| OpenVLA -OFT | FP Model | - | 97.6% | 98.4% | 97.9% | 94.5% | 97.1% | - | 15.4 | 1× |
| | W8A8 | SmoothQuant | 96.4% | 97.8% | 95.4% | 94.3% | 96.0% | -1.1% | 7.7 | 1.41× |
| | | OmniQuant | 95.4% | 96.2% | 93.0% | 92.6% | 94.3% | -2.8% | 8.0 | 1.30× |
| | | QVLA | 97.2% | 98.2% | 95.8% | 94.3% | 96.4% | -0.7% | 7.2 | 1.36× |
| | W4A4 | SmoothQuant | 77.2% | 70.0% | 77.8% | 68.6% | 73.4% | -23.7% | 4.9 | 1.53× |
| | | OmniQuant | 95.0% | 94.4% | 94.0% | 92.0% | 93.9% | -3.2% | 5.7 | 1.37× |
| | | QVLA | 96.2% | 97.6% | 96.4% | 93.8% | 96.0% | -1.1% | 4.5 | 1.49× |

$b_{l,c} \in \{0, 2, 4, 8, 16\}$ to each channel to minimize the action error, subject to an average budget $\bar{B}$:

$$\min_{\{b_{l,c}\}} \sum_{l,c} s_{l,c}^{(b_{l,c})} \quad \text{s.t.} \quad \frac{1}{N} \sum_{l,c} b_{l,c} \leq \bar{B}, \tag{7}$$

where $N$ is the total number of channels and 0-bit signifies pruning. To solve this NP-hard problem efficiently, we propose a greedy demotion algorithm. The procedure begins by initializing all channels to the highest precision, 16-bit. It then proceeds in a series of demotion stages ($16 \rightarrow 8$, $8 \rightarrow 4$, $4 \rightarrow 2$, and $2 \rightarrow 0$) until the average bit budget $\bar{B}$ is met. Within each stage, from a higher bit-width $b_{\text{hi}}$ to a lower one $b_{\text{lo}}$, we evaluate the cost-effectiveness of demoting each candidate channel using the sensitivity-to-bit ratio $\rho_{l,c}$:

$$\rho_{l,c} = \frac{s_{l,c}^{(b_{\text{lo}})} - s_{l,c}^{(b_{\text{hi}})}}{b_{\text{hi}} - b_{\text{lo}}}. \tag{8}$$

This ratio represents the marginal increase in error for each bit saved. To make the most efficient bit reductions, we prioritize demoting channels that are least sensitive to quantization, *i.e.*, those with the lowest $\rho_{l,c}$ values. Specifically, in the $16 \rightarrow 8$ stage, we sort all channels by their corresponding $\rho_{l,c}$ in ascending order and sequentially demote them to 8-bit. After each demotion, we check the total bit budget. If the budget $\bar{B}$ is met, the process stops. Otherwise, the algorithm proceeds to the $8 \rightarrow 4$ stage, repeating the sort-and-demote process for all channels currently at 8-bit, and so on for the remaining stages. To further refine the final bit assignment, we introduce several heuristics. To mitigate over-pruning, the final $2 \rightarrow 0$ demotion stage is regularized using dual-threshold and L0-style constraints. The computational complexity of this method is dominated by sorting, scaling as $\mathcal{O}(C \log C)$, where $C$ is the total number of channels.

Activations are assigned a uniform bit-width (*e.g.*, 8-bit) using distribution-aware calibration, which ensures a *branch-free* execution path and stable latency. Weights are stored on a per-row basis, each with its own scale and zero-point, and are dequantized upon access. This approach avoids per-channel runtime branching. The scheme applies to all linear and convolutional layers in both the vision and language backbones. See Appendix for more detailed pseudocode.

## 4 EXPERIMENTS

### 4.1 EXPERIMENTAL SETTINGS AND DETAILS

To evaluate our method, we adopt the widely used LIBERO benchmark (Liu et al., 2024a), comprising four distinct task suites for robot manipulation. Our floating-point (FP) baseline employs models with weights in the BF16 format, and all experiments were conducted on NVIDIA RTX 4090 GPUs. In our quantization strategy, we selectively apply channel-wise weight quantization with gated bit

Table 2: **Performances under various weight-only quantization settings**. Weight-only quantization primarily targets the reduction of memory, typically offering a marginal improvement in latency.

| Model | Setting | Method | Spatial | Object | Goal | Long | Avg. ↑ | Δ | Mem. (GB) ↓ |
|---|---|---|---|---|---|---|---|---|---|
| OpenVLA | FP Model | – | 84.7% | 88.4% | 79.2% | 53.7% | 76.5% | – | 15.2 |
| | W8A16 | AWQ | 82.3% | 87.2% | 74.6% | 50.7% | 73.7% | -1.8% | 7.6 |
| | | QVLA | 86.2% | 88.4% | 79.4% | 53.1% | 76.8% | +0.3% | 7.2 |
| | W4A16 | AWQ | 80.0% | 81.2% | 74.6% | 47.2% | 70.8% | -4.7% | 5.0 |
| | | QVLA | 86.0% | 88.6% | 78.4% | 52.8% | 76.5% | +0.0% | 4.3 |
| OpenVLA-OFT | FP Model | – | 97.6% | 98.4% | 97.9% | 94.5% | 97.1% | – | 15.4 |
| | W8A16 | AWQ | 95.2% | 96.8% | 95.4% | 93.1% | 95.1% | -2.0% | 8.0 |
| | | QVLA | 97.4% | 98.6% | 97.2% | 94.6% | 97.0% | -0.1% | 7.4 |
| | W4A16 | AWQ | 93.0% | 92.4% | 93.8% | 90.7% | 92.5% | -4.5% | 5.2 |
| | | QVLA | 97.0% | 98.4% | 96.8% | 94.4% | 96.7% | -0.4% | 4.5 |

allocation to the vision backbone and language module. The projector and action head remain in full BF16 precision to preserve control stability. The core of our method is an action-centric sensitivity analysis, which requires a calibration set sampled from LIBERO training demonstrations and augmented with a small instruction-only subset. Using this set, we measure the sensitivity score (Eq. 4) for each channel in the target modules by simulating its quantization to various bit-widths $b \in \{0, 2, 4, 8, 16\}$. To confirm the validity of this metric, the resulting sensitivity rankings are further cross-validated with short environmental rollouts. While our diagnostic analysis spans all modules, the final average-bit budget and the effects of pruning (0-bit quantization) are computed exclusively over the layers designated for quantization, ensuring a fair comparison.

## 4.2 COMPARISON WITH STATE-OF-THE-ARTS

**Results on weight-activation quantization.** Tab. 1 compares the performance of our QVLA with SmoothQuant Xiao et al. (2022), OmniQuant Shao et al. (2024), two methods initially designed for LLMs/VLLMs, under weight-activation quantization. The results demonstrate that QVLA achieves a superior trade-off between high task success rates, low memory footprint, and fast inference speed. For instance, under the W4A4 quantization setting for the OpenVLA model, QVLA retains **99.3%** of the full-precision performance, incurring a minimal accuracy drop of only **0.5%**. This is substantially smaller than the 13.3% drop from SmoothQuant and the 3.2% from OmniQuant. This comprehensive advantage is achieved while requiring only **28.2%** of the original model's memory and delivering a **1.47×** inference speedup. These results validate the effectiveness of our action-centric quantization strategy, establishing QVLA as a compelling choice for balancing performance and efficiency in resource-constrained environments.

**Results on weight-only quantization.** The performance comparison under weight-only quantization, presented in Tab. 2, reveals the consistent superiority of our QVLA. This advantage is particularly pronounced in the most challenging W4A16 setting. Specifically, on the OpenVLA model, QVLA incurred **zero performance loss**, whereas AWQ's Lin et al. (2023) average success rate dropped by 4.7%. Similarly, for the OpenVLA-OFT model, QVLA's performance degradation was minimal at just **0.1%** and **0.4%**, markedly lower than the 2.0% and 4.5% losses observed with AWQ. Collectively, these results robustly demonstrate our QVLA's superior ability to preserve model accuracy under aggressive compression.

## 4.3 FURTHER ANALYSIS

**The superiority of channel-wise quantization.** We conducted an ablation study with Open-VLA Kim et al. (2024) to validate our choice of channel-wise quantization over the conventional layer-wise method for the VLA model. As detailed in Tab. 3, our approach demonstrates a distinct advantage when benchmarked against the 76.5% success rate of the full-precision baseline on LIBERO. At INT4 precision, the channel-wise method **perfectly preserves** the baseline performance (76.5%), whereas its layer-wise counterpart suffers a notable decline to 74.8%. This superi-

Table 3: Comparison of layer-wise and channel-wise quantization methods on various tasks. The baseline is OpenVLA.

| Weight Type | Method | | Spatial | Object | Goal | Long | Avg. ↑ |
| --- | --- | --- | --- | --- | --- | --- | --- |
| | Layer | Channel | | | | | |
| FP Model | – | – | 84.7% | 88.4% | 79.2% | 53.7% | 76.5% |
| INT4 | ✓ | | 84.2% | 86.8% | 77.0% | 51.2% | 74.8% |
| | | ✓ | 86.0% | 88.6% | 78.4% | 52.8% | 76.5% |
| INT8 | ✓ | | 84.0% | 87.2% | 77.8% | 50.6% | 74.9% |
| | | ✓ | 86.2% | 88.4% | 79.4% | 53.1% | 76.8% |

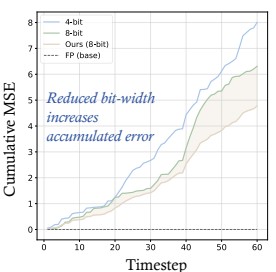

Figure 3: Quantification of temporal error accumulation.

Table 4: **The influence of pruning (0-bit quantization) and uniform-bit quantization under an overall INT8 budget.** The row ④ with "prune" and without "uniform-bit" indicates our proposed quantization method with the $\{0, 2, 4, 8, 16\}$ as candidates during assigning bit to each channel.

| # | Weight Type | Quantization Method | | Spatial | Object | Goal | Long | Avg. ↑ | Memory (GB) |
| --- | --- | --- | --- | --- | --- | --- | --- | --- | --- |
| | | Prune | Uniform Bit | | | | | | |
| ① | FP Model | – | – | 84.7% | 88.4% | 79.2% | 53.7% | 76.5% | 15.2 |
| ② | INT8 | | | 86.4% | 88.0% | 79.0% | 53.4% | 76.7% | 7.5 |
| ③ | | | ✓ | 83.6% | 87.0% | 77.4% | 50.2% | 74.6% | 7.6 |
| ④ | | ✓ | | 86.2% | 88.4% | 79.4% | 53.1% | 76.8% | 7.0 |
| ⑤ | | ✓ | ✓ | 84.0% | 87.2% | 77.0% | 50.4% | 74.7% | 7.1 |

ority is even more pronounced at INT8 precision, where our method not only matches but **surpasses** the baseline at 76.8%. In stark contrast, the layer-wise approach again degrades performance, falling to 74.9%. This study provides compelling evidence that channel-wise quantization is the superior strategy for compressing our VLA model.

**Mitigating temporal error accumulation.** Fig. 3 illustrates the temporal accumulation of action errors. As expected, cumulative error increases over time for all quantization methods, with 4-bit quantization showing a significantly faster rate of error growth than 8-bit methods. Crucially, our proposed 8-bit method consistently maintains a lower cumulative error than the uniform 8-bit baseline. This performance gap widens progressively over the time horizon, highlighting our method's superior ability to mitigate long-horizon error propagation. This sustained reduction in error demonstrates the enhanced stability and long-horizon robustness conferred by our approach, an advantage that becomes particularly pronounced as dynamic effects accumulate over longer episodes.

**The impact of prune and uniform bit.** As shown in Tab. 4, channel-wise gating with $\{2, 4, 8, 16\}$ bits as candidates for each channel's quantization already matches the full-precision (FP) baseline performance (② 76.7% *vs.* ① 76.5%), confirming the effectiveness of concentrating bits on critical channels. When combined with pruning (from ②$\{2, 4, 8, 16\}$ to ④$\{0, 2, 4, 8, 16\}$), this method further reduces memory to 7.0 GB while also slightly boosting the average success rate to 76.8%. In contrast, enforcing a uniform bit-width (③ only 8-bit) causes a substantial drop in performance to 74.6%. Subsequent pruning (⑤$\{0, 8\}$) is unable to recover this loss, yielding only 74.7%. This indicates that a uniform quantization strategy is fundamentally ill-suited for the VLA model. Therefore, under an overall INT8 budget, the combination of channel-wise gating with pruning (0-bit) proves to be the superior strategy, achieving the highest accuracy while minimizing the memory footprint.

## 4.4 PERFORMANCE ON REAL-WORLD TASKS

**Real-world Setup.** We construct a bimanual robotic system to perform real-world manipulation tasks. Two IMETA-Y1 robotic arms are employed for execution. The system utilizes three Orbbec DaBai DCW2 cameras to capture visual observations: two are mounted on the wrists to provide egocentric views, and one is fixed externally to capture the global third-person view. The real-world robotic system is shown in Fig. 4.

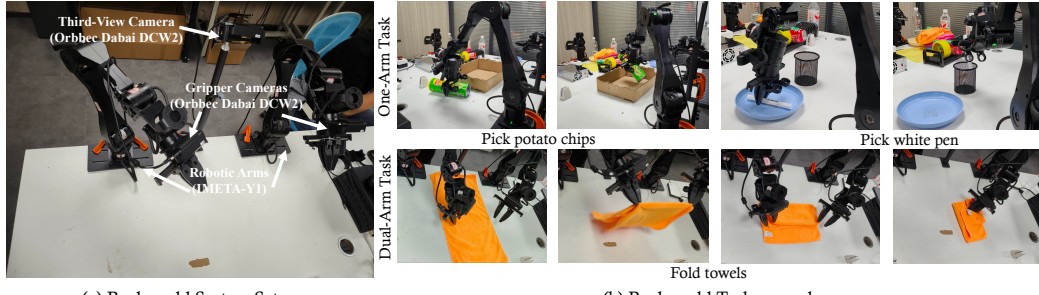

(a) Real-world System Setup        (b) Real-world Task examples

Figure 4: Real-world system IMETA-Y1 and the task examples. The system consists of two IMETA-Y1 robotic arms and three Orbbec DaBai DCW2 cameras.

Table 5: Real-world success rate comparison with QVLA.

| Method | Setting | One-Arm Task | | Dual-Arm Task | Average | SpeedUp ↑ |
|---|---|---|---|---|---|---|
| | | Pick white pen | Pick potato chips | Fold towels | | |
| $\pi_0$ | - | 8/10 | 7/10 | 4/10 | 63.3% | 1.00× |
| QVLA | W8A16 | 8/10 | 6/10 | 5/10 | 63.3% | 1.28× |

**Tasks and Datasets.** Our dataset cover both single-arm and dual-arm tasks, as well as simple and complex dexterous tasks. Simple tasks are short-horizon operations, such as placing a white pen into a pen holder or grasping and transferring potato chips into a bin. Dexterous tasks require continuous and sustained control, such as folding a towel which typically involve fine-grained manipulation of deformable objects. We collected fifty trajectories for each single-arm task and one hundred trajectories for the dual-arm tasks, all sampled at a frequency of 30Hz.

**Results.** We use the $\pi_0$ model as a real-world baseline and performed quantization under the W8A16 setting. Tab. 5 presents the results of the real-world experiments. We observe that QVLA performs excellently on both single-arm and dual-arm tasks, with performance essentially on par with the original model. This strongly demonstrates the robustness of our quantization method, which maintains good performance even in real-world environments. Furthermore, we achieved an acceleration ratio of $1.28\times$ under the W8A16 setting. Our model is trained on NVIDIA A100 GPUs and deployed for inference on a single NVIDIA 4070 GPU.

For additional technical depth, please refer to the Appendix, including the rigorous theoretical derivation of our sensitivity proxy and the complete algorithmic details. The Appendix also presents generalization results on the UniVLA architecture and CALVIN benchmark, alongside qualitative policy visualizations and ablation studies that further substantiate the robustness and architectural transferability of our framework.

# 5 CONCLUSION

This paper presents the first systematic analysis of quantization challenges in VLA models. We demonstrate that naively migrating quantization strategies from other domains, such as the uniform-bit methods common for LLMs/MLLMs, is ill-suited for these action-driven models and leads to significant performance degradation. This failure stems from the VLA's unique sensitivity to noise, a critical barrier to their deployment on resource-constrained robots. To address this gap, we propose QVLA, an adaptive quantization framework specifically designed for VLAs. Our action-centric approach reframes the problem by anchoring optimization objectives in the action space. It unifies quantization and pruning via a per-channel sensitivity metric and a global greedy algorithm. Extensive evaluations on OpenVLA and OpenVLA-OFT baselines confirm that QVLA outperforms conventional methods, reducing action errors and enhancing task success rates to a level that can even surpass the full-precision baseline. Looking ahead, we are going to adapt QVLA to a broader spectrum of VLA architectures. This further validate our framework's generalizability and solidify its role as a key enabler for deploying capable foundation models on real-world robotic systems.

**Acknowledgement**. This work was supported in part by the Natural Science Foundation of China (Grant No.62503323, 62372451, 62192785, 62372082), the Beijing Natural Science Foundation (JQ24022), CAAI-Ant Group Research Fund (CAAI-MYJJ 2024-02), Young Elite Scientists Sponsorship Program by CAST (2024QNRC001).

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
