## A  THE USE OF LARGE LANGUAGE MODELS (LLMs)

The authors utilized LLMs such as Google's Gemini to refine the language and enhance the readability of this paper. It is important to state that these models were not used for generating ideas or the conceptual framework.

## B  ALGORITHM

---

**Algorithm 1:** QVLA: Greedy Bit Allocation under an Average Bit Budget

---

**Input:** Calibration set $\mathcal{D}$; per-channel action-space sensitivities $s_{l,c}^{(b)}$ for $b \in \{16, 8, 4, 2, 0\}$; target average bit-width $\bar{B}$

**Output:** Assigned bit-widths $b_{l,c} \in \{16, 8, 4, 2, 0\}$ for all output channels $(l, c)$

1 **Init:** For all $(l, c)$, set $b_{l,c} \leftarrow 16$; let $N \leftarrow$ number of channels; $S_{\text{init}} \leftarrow 16 \cdot N$; $S_{\text{target}} \leftarrow \bar{B} \cdot N$; $\Delta S \leftarrow S_{\text{init}} - S_{\text{target}}$. Create a min-heap $H$ keyed by $\rho$.;

2 **Push first-step (16→8): foreach** *channel* $(l, c)$ **do**
3      push $\big((l, c), (16{\to}8)\big)$ into $H$ with key $\rho_{l,c}(16{\to}8)$;

4 **while** $\Delta S > 0$ *and* $H$ *not empty* **do**
5      $\big((l, c), (h{\to}l)\big) \leftarrow \text{pop\_min}(H)$ ;       // smallest loss per saved bit
6      **if** $b_{l,c} = h$ **then**
7           $b_{l,c} \leftarrow l$;   $\Delta S \leftarrow \Delta S - \text{save}(h{\to}l)$;
          // enqueue the next adjacent step if any: 16→8 then 8→4, etc.
8           **if** $(l{\to}l') \in \{8{\to}4, 4{\to}2, 2{\to}0\}$ **then**
9               push $\big((l, c), (l{\to}l')\big)$ with key $\rho_{l,c}(l{\to}l')$ into $H$;

10 **return** $\{b_{l,c}\}$;

11 **Remarks:** Adjacent-only demotions (16→8→4→2→0). 0-bit means pruning.

---

## C  IMPACT OF THE GATES RATIO

**Impact of the gates ratio.** The gate ratio on model performance while maintaining an overall INT8 budget. The results, summarized in Tab. 6, reveal a clear trend that optimal performance (76.3% success rate) is achieved when a high proportion of parameters are quantized to 8-bit. Conversely, performance systematically degrades as this ratio is reduced. In future work, we will develop an automated algorithm to dynamically determine the optimal gate ratio for any given budget, thus eliminating the need for heuristic tuning.

Table 6: **Performance results of OpenVLA with different gates ratios under the INT8 budget.** We compare the performance across various gate ratios based on their success rates.

| Budget | Gates Ratio (0bit:2bit:4bit:8bit:16bit) | LIBERO-Spatial | LIBERO-Object | LIBERO-Goal | LIBERO-Long | Average |
|---|---|---|---|---|---|---|
| INT8 | 1% : 5% : 22% : 56% : 16% | 84.6% | 88.2% | 79.0% | 53.2% | 76.3% |
| | 5% : 8% : 27% : 35.5% : 24.5% | 84.0% | 87.4% | 78.0% | 53.0% | 75.6% |
| | 8% : 10% : 32% : 18.5% : 31.5% | 83.2% | 85.8% | 77.4% | 52.0% | 74.6% |
| | 10% : 12% : 22% : 25% : 31% | 82.7% | 85.2% | 76.0% | 52.4% | 74.1% |

## D  EXPANDED EVALUATION ON UNIVLA

To robustly demonstrate the generalizability and architectural transferability of AutoQVLA, we extended our evaluation to the **UniVLA-7B** model, a VLA architecture notable for its distinct task-

Table 7: Performance comparison on the **UniVLA-7B** model under various quantization settings.

| Model | Setting | Method | Spatial | Object | Goal | Long | Avg. | Mem. (GB) ↓ |
|---|---|---|---|---|---|---|---|---|
| | BF16 | - | 96.5% | 96.8% | 95.6% | 92.0% | 95.2% | 14.6 |
| | W8A16 | AWQ | 94.4% | 95.6% | 94.8% | 91.0% | 94.0% | 7.8 |
| | | **AutoQVLA** | 97.2% | 96.6% | 95.8% | 91.6% | 95.3% | 7.4 |
| UniVLA-7B | W4A16 | AWQ | 92.6% | 94.2% | 94.0% | 89.8% | 92.6% | 5.2 |
| | | **AutoQVLA** | 96.8% | 97.2% | 95.0% | 91.4% | 95.1% | 4.7 |
| | W8A8 | SmoothQuant | 96.0% | 96.4% | 93.2% | 91.2% | 94.2% | 7.2 |
| | | OmniQuant | 94.0% | 94.8% | 93.6% | 90.4% | 93.2% | 7.5 |
| | | **AutoQVLA** | 96.8% | 96.6% | 94.2% | 91.0% | 94.7% | 6.9 |

Table 8: Performance evaluation on the **CALVIN** benchmark: Task Success Rate by Sequence Length. The compressed models are evaluated under the W8A16 setting.

| Method | Task completed in a row (%) | | | | | Avg. len ↑ |
|---|---|---|---|---|---|---|
| | 1 | 2 | 3 | 4 | 5 | |
| OpenVLA-OFT (FP) | 96.3 | 89.1 | 82.4 | 75.8 | 66.5 | 4.10 |
| AWQ | 95.7 | 87.5 | 76.8 | 69.5 | 64.3 | 3.97 |
| **AutoQVLA** | 95.9 | 88.6 | 81.4 | 72.1 | 63.2 | 4.03 |

centric latent action decoding strategy. The results, summarized in Tab. 7, confirm that the core principle of action-centric channel sensitivity effectively generalizes to VLA models with diverse internal structures.

As detailed in Tab. 7, our AutoQVLA consistently surpasses leading quantization methods, including GPTQ, AWQ, SmoothQuant, and OmniQuant across various low-bit configurations on the LIBERO benchmark. For the aggressively compressed W4A16 setting, for example, our method achieved an average success rate of 95.1%, a significant gain over the 92.6% achieved by AWQ. Crucially, our approach also produces the most compact models, requiring the lowest memory footprint among all tested quantization methods. The advantages in both performance and memory efficiency demonstrates that the core principle of action-centric channel sensitivity generalizes effectively to VLA models with diverse internal structures.

# E    EVALUATION ON THE CALVIN BENCHMARK

To validate the robustness of our framework against increased task complexity and dynamic environments, we extended our evaluation to the **CALVIN** benchmark. CALVIN presents more intricate sequence planning and interaction challenges compared to LIBERO. On this benchmark, the OpenVLA-OFT model compressed with our AutoQVLA (W8A16) exhibited negligible performance degradation, successfully preserving long-horizon task success and sequence stability. This finding reinforces our core argument that minimizing action-space error through adaptive bit allocation is essential for maintaining robust policy behavior across diverse and challenging robotics applications.

# F    CALIBRATION SET DETAILS AND SIZE ABLATION

The calibration set $\mathcal{D}$ for the LIBERO is curated directly from the official LIBERO training trajectories. We randomly sampled a total of 512 trajectories from the combined task training data to serve as the calibration set for our sensitivity analysis and bit allocation process. This dataset is specifically utilized for the action-space sensitivity analysis outlined in Eq. 5.

To evaluate the impact of the calibration set size on performance, we conducted an ablation study using OpenVLA-OFT (W8A16). As presented in Tab.9, a set size of 512 yields the highest average success rate (97.0%). Crucially, the performance remains highly stable with neighboring set sizes: using 256 or 1024 trajectories results in only a marginal decrease in performance (96.8% and 96.7%, respectively). This demonstrates the robustness of our AutoQVLA.

Table 9: Ablation Study on Calibration Set Size. Evaluation on OpenVLA-OFT (W8A16) across LIBERO tasks.

| Calibration Set Size | Spatial | Object | Goal | Long | Avg. |
|---|---|---|---|---|---|
| 128 | 97.0 | 97.8 | 96.8 | 94.0 | 96.4 |
| 256 | 97.2 | 98.2 | 97.0 | 94.6 | 96.8 |
| 512 | 97.4 | 98.6 | 97.2 | 94.6 | 97.0 |
| 1024 | 97.0 | 98.4 | 97.0 | 94.4 | 96.7 |

# G  THEORETICAL ANALYSIS OF ACTION-SPACE SENSITIVITY APPROXIMATION

A rigorous theoretical foundation is indeed crucial for justifying our design choices. We have incorporated detailed theoretical analysis into the revised manuscript, providing the derivation for our first-order proxy used to efficiently estimate the action-space sensitivity.

The true sensitivity we aim to minimize is the action space Mean Squared Error (MSE) when quantizing layer $l$, channel $c$ to bit-width $b$:

$$S_{l,c}^{(b)} := \mathbb{E}\big[\|\Delta A\|^2\big], \quad \Delta A = A(X_{l,c} + \Delta X_{l,c}^{(b)}) - A(X_{l,c}),$$

where $A(\cdot)$ is the mapping from the channel output to the final action, and $\Delta X_{l,c}^{(b)}$ is the quantization noise.

Assuming a local linearity for small perturbations, we apply a first-order Taylor expansion to approximate the action deviation:

$$\Delta A \approx J_{A,X_{l,c}} \Delta X_{l,c}^{(b)},$$

where $J_{A,X_{l,c}}$ is the Jacobian of the action $A$ with respect to the channel output $X_{l,c}$. Substituting this approximation into $S_{l,c}^{(b)}$ yields:

$$S_{l,c}^{(b)} \approx \mathbb{E}\big[\|J_{A,X_{l,c}}\Delta X_{l,c}^{(b)}\|^2\big] = \mathbb{E}\Big[(\Delta X_{l,c}^{(b)})^\top \underbrace{(J_{A,X_{l,c}}^\top J_{A,X_{l,c}})}_{=:H_{l,c}} \Delta X_{l,c}^{(b)}\Big].$$

We then employ the common uniform quantization noise model, characterized by zero mean and an isotropic covariance structure:

$$\mathbb{E}[\Delta X_{l,c}^{(b)}] = 0, \quad \mathrm{Cov}(\Delta X_{l,c}^{(b)}) = (\sigma_{l,c}^{(b)})^2 I.$$

Under this assumption, the expected quadratic form simplifies via the matrix trace property $\mathbb{E}[\mathbf{x}^\top H \mathbf{x}] = \mathrm{Tr}(H\mathrm{Cov}(\mathbf{x}))$:

$$\mathbb{E}\big[(\Delta X_{l,c}^{(b)})^\top H_{l,c} \Delta X_{l,c}^{(b)}\big] = \mathrm{Tr}\big(H_{l,c} \mathrm{Cov}(\Delta X_{l,c}^{(b)})\big) = (\sigma_{l,c}^{(b)})^2 \mathrm{Tr}(H_{l,c}).$$

Since $\mathrm{Tr}(H_{l,c}) = \mathrm{Tr}(J_{A,X_{l,c}}^\top J_{A,X_{l,c}}) = \|J_{A,X_{l,c}}\|_F^2$ (the squared Frobenius norm of the Jacobian), we arrive at the final first-order approximation:

$$S_{l,c}^{(b)} \approx (\sigma_{l,c}^{(b)})^2 \cdot \|J_{A,X_{l,c}}\|_F^2.$$

This theoretical derivation rigorously justifies our efficient proxy, which factors the sensitivity into a predictable quantization error term $(\sigma_{l,c}^{(b)})^2$ and a task-dependent local sensitivity gain ($\|J_{A,X_{l,c}}\|_F^2$).

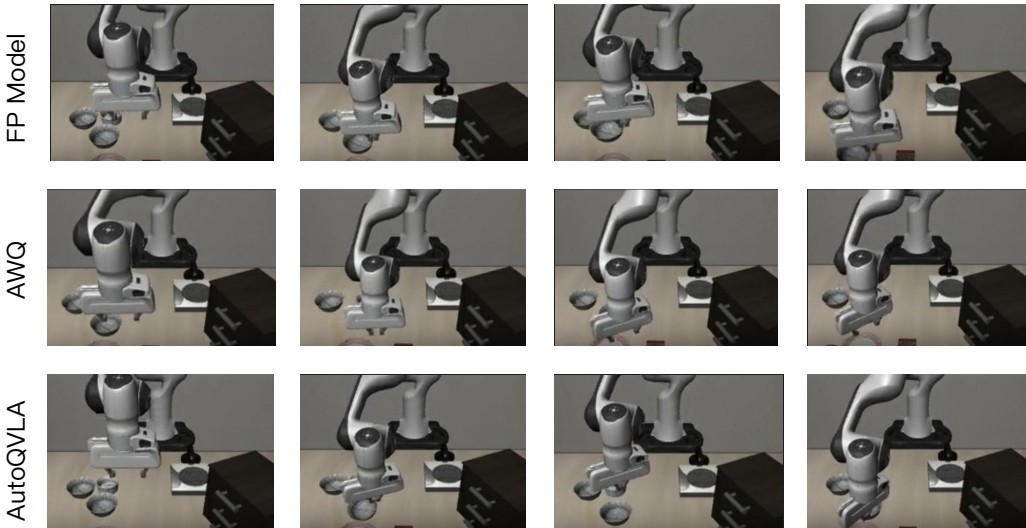

Figure 5: Rollouts from LIBERO. The task is "Pick up the black bowl between the plate and the small bowl and put it on the plate".

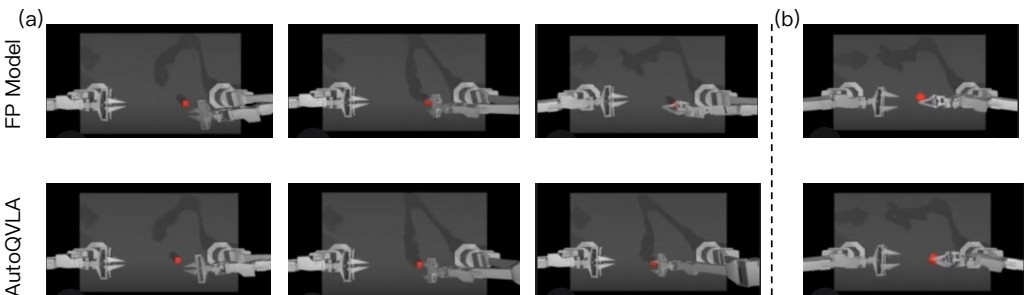

Figure 6: Rollouts from ALOHA. The task is "transfer the cube", (a) and (b) are from different seeds.

# H QUALITATIVE ANALYSIS OF QUANTIZED POLICY BEHAVIOR

To provide critical insight beyond scalar success rates we performed a qualitative analysis by inspecting policy rollouts from the LIBERO and ALOHA evaluation benchmarks. This analysis highlights how the action-centric quantization of AutoQVLA affects the policy's fine-grained control compared to full-precision (FP) models and methods derived from uniform LLM quantization.

## H.1 BEHAVIOR DURING FULL-PRECISION SUCCESS TRIALS

In instances where the full-precision (FP) model achieves task completion, the differentiating factor among quantization schemes lies primarily in the precision and stability of the executed actions. Conversely, failures in competing quantized models are almost always concentrated during crucial manipulation phases like initial grasping, establishing stable contact, or final object placement. Approaches prioritizing passive fidelity, such as AWQ, frequently show marked inaccuracies during these key moments. For example, when attempting to grasp an object, the quantized gripper may significantly overshoot the target (as shown in Fig. 5). Alternatively, during the final goal-conditioned placement, the gripper might prematurely release the object, resulting in an evaluated failure even if the high-level planning was correct. The superior performance of AutoQVLA stems directly from its capability to maintain higher action precision throughout these decisive control steps, a mechanism that effectively dampens the accumulation of errors that lead to catastrophic failures in alternative quantization methods.

## H.2 BEHAVIOR DURING FULL-PRECISION FAILURE TRIALS

We also investigated policy behavior in scenarios where the FP model failed the task for example due to initial miss or error accumulation. Quantized models generally demonstrate a slightly weaker capacity for robust re-localization compared to the FP baseline. If an initial grasp fails or a distant object is missed the FP model typically proceeds with a reasonable fallback motion in contrast, certain quantized models, especially when attempting to recover a distant object may exhibit localized oscillation or jerkiness without initiating a clear effective re-engagement trajectory, as shown in Fig. 6 (a). Interestingly in specific complex grasping instances within the ALOHA benchmark, the moderate quantization noise introduced by AutoQVLA appeared to have a regularization effect on the policy. This occasionally resulted in the quantized policy producing demonstrably cleaner and more effective contact configurations for grasping than the full-precision baseline(As shown in Fig. 6 (b)).

These qualitative observations affirm that the quantitative superiority of AutoQVLA stems from its enhanced ability to preserve high-fidelity control signals precisely where minor deviations would lead to compounding physical errors.

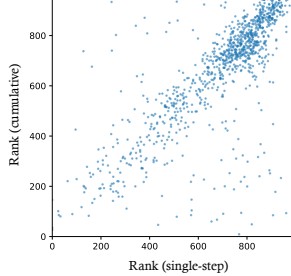

Figure 7: Rank scatter of single-step vs. cumulative sensitivity.

## I THE RANKINGS OF CHANNEL SENSITIVITIES

We randomly sample 1,000 channels and compare the rankings produced by the cumulative and single-step sensitivity metrics. Each point in the scatter plot corresponds to one channel. when a point lies on the diagonal, it means that this channel receives the same rank under both metrics. As shown in Fig. 7, roughly 80% of the points cluster closely around the diagonal, indicating that the two rankings are highly consistent.