# OpenReview forum: "QVLA: Not All Channels Are Equal in Vision-Language-Action Model's Quantization"
_ICLR.cc/2026/Conference — ICLR 2026 Poster_

### Official Review · Reviewer_RtD1 · 2025-10-31

**Soundness:** 4
**Presentation:** 3
**Contribution:** 4
**Rating:** 6
**Confidence:** 4

**Summary:**

The paper proposes AutoQVLA, a VLA quantization method that reduces memory requirements and inference latency of VLAs by employing channel-wise bit allocation and pruning which optimize for reduced degradation of action quality. The authors show that naive uniform-bit quantization methods used for LLMs underperforms when applied to VLAs since they do not take into account how errors compound across trajectories when the VLAs are deployed as policies. The authors propose a more sophisticated framework that first analyzes the action sensitivity of different channels within network layers and then runs a greedy demotion algorithm to iteratively demote channels from full precision to lower precision until a target bit budget is met. Experimental results with OpenVLA and OpenVLA-OFT policies in the LIBERO simulation benchmark demonstrate nearly 4x VRAM reduction and 1.5x faster inference speed, with only up to 1% absolute drop in average success rate, outperforming prior quantization methods.

**Strengths:**

* The paper is well motivated and generally easy to follow.
* The paper achieves significant reduction in memory requirements (about 4x) and improvement in inference latency (about 1.5x) while preserving most of the accuracy of full-precision VLAs. These efficiency results surpass those of prior methods (SmoothQuant, OmniQuant, AWQ), and the degradation in policy performance is substantially lower. This suggests that the method can be practically useful for low-resource deployment of VLA policies.
* There is good breadth in the experiments: The analyses cover weights-and-activations and only-weights quantization, layer- vs channel-wise quantization, temporal error accumulation across different quantization schemes, and 0-bit ("pruning") and uniform-bit quantization.
* The analysis of quantization is performed on two VLA instances that vary substantially in their action generation schemes (OpenVLA which does vanilla autoregressive decoding of tokens and OpenVLA-OFT which does parallel decoding of continuous actions). Findings hold across both instances and overall support the authors' claims.

**Weaknesses:**

Major:
* The experiments are limited to one benchmark of robot tasks in simulation (LIBERO). Results in another simulator (e.g. RoboCasa or CALVIN) or real-world robotic manipulation tasks would further strengthen the paper.
* The temporal error accumulation analysis in Section 4.3 and Figure 3 could benefit from greater detail. It is unclear under which evaluation setting the curves in the plot are computed (e.g. what is the task here, what is the policy being analyzed?). In addition, some discussion on how the differences in cumulative MSE affect the episode outcomes can be helpful (e.g. what do these differences mean in terms of success rate)?
* Qualitative analysis of how the AutoQVLA-quantized VLA policies behave in comparison to the full-precision policies or the policies quantized using prior methods is missing and can further strengthen the paper. I.e., beyond success rate differences, how do the AutoQVLA policies actually behave when rolled out as policies, and how do they succeed/fail? Are there example rollouts that can illustrate why the other quantization approaches are less effective?
* In Section 3.3.1, the authors write, "In practice, we find that the rankings of channel sensitivities produced by $s_{l,c}^{(b)}$ and $S_{l,c}^{(b)}$ are highly consistent. This crucial finding allows us to leverage the computationally cheaper single-step metric $s_{l,c}^{(b)}$ for guiding the bit allocation process, while using the more comprehensive cumulative metric $S_{l,c}^{(b)}$ to validate that our approach successfully extrapolates to long-horizon performance." This finding is important, yet evidence supporting it is missing. Details on how this conclusion was made would be helpful.
* There is lack of detail on the calibration set used for LIBERO and how it is curated.
* It is unclear how many trials/random seeds are used for the evaluations (e.g. Table 1 and 2).

Minor:
* Figure 1(b): The separation of vision encoder, projector, language model, and action head is not very clear. For example, where does the language model section start exactly in this plot?
* Figure 2 step 1: "tagrt bit" typo?
* Table 4: "The row (3)" typo? Shouldn't it be "The row (4)" instead?

**Questions:**

See questions in Weaknesses section.

---

> ### Author Response · Authors · 2025-11-23
> **(Part 1) Response to Reviewer RtD1**
>
> ### W1: About Generalizability and Scope of Evaluation
>
>
>
>
> Thanks for your question. To further establish the robustness and generalizability of our AutoQVLA, we extend the evaluation to a broader set of baselines, benchmarks and real-world experiments.
>
> **1. Generalization to Different VLA architectures (UniVLA)**
>
>
> We extended our evaluation to UniVLA-7B, a model notable for its distinct task-centric latent action decoding. As detailed in the following table, our AutoQVLA consistently surpasses leading quantization methods including AWQ, SmoothQuant, and OmniQuant across various low-bit configurations on the LIBERO benchmark. For W4A16 setting, for example, our method achieved an average success rate of 95.10%, a significant gain over the 92.65% from AWQ. Crucially, our approach also produces the most compact models, requiring the lowest memory footprint among all tested quantization methods. The advantages in both performance and memory efficiency demonstrates that the core principle of action-centric channel sensitivity generalizes effectively to VLA models with diverse internal structures.
>
> | Model | Setting | Method | Spatial | Object | Goal | Long | Ave. | Memory |
> | :--- | :--- | :--- | :--- | :--- | :--- | :--- | :--- | :--- |
> | univla-7b | BF16 | - | 96.5% | 96.8% | 95.6% | 92.0% | 95.2% | 14.6GB |
> |Weight-Only | W8A16 | AWQ | 94.4% | 95.6% | 94.8% | 91.0% | 94.0% | 7.8GB |
> | | W8A16 | AutoQVLA (Ours) | 97.2% | 96.6% | 95.8% | 91.6% | 95.3% | 7.4GB |
> | | W4A16 | AWQ | 92.6% | 94.2% | 94.0% | 89.8% | 92.6% | 5.2GB |
> | | W4A16 | AutoQVLA (Ours) | 96.8% | 97.2% | 95.0% | 91.4% | 95.1% | 4.7GB |
> |Weight-Activation | W8A8 | Smoothquant | 96.0% | 96.4% | 93.2% | 91.2% | 94.2% | 7.2GB |
> | | W8A8 | Omniquant | 94.0% | 94.8% | 93.6% | 90.4% | 93.2% | 7.5GB |
> | | W8A8 | AutoQVLA (Ours) | 96.8% | 96.6% | 94.2% | 91.0% | 94.7% | 6.9GB |
>
> **2. Generalization to Different Tasks and Benchmarks (CALVIN)**
>
>
> To validate the robustness of our framework against increased task complexity and dynamic environments, we extended our evaluation to the CALVIN benchmark. CALVIN presents more intricate sequence planning and interaction challenges compared to LIBERO. On this benchmark, the OpenVLA-OFT model compressed with our AutoQVLA (W8A16) exhibited negligible performance degradation, successfully preserving long-horizon task success and sequence stability. This finding reinforces our core argument that minimizing action-space error through adaptive bit allocation is essential for maintaining robust policy behavior across diverse and challenging robotics applications.
>
> | CALVIN ABC->D | Task completed in a row | | | | | Avg.len↑ |
> | :--- | :--- | :--- | :--- | :--- | :--- | :--- |
> | Method| 1 | 2 | 3 | 4 | 5 | |
> | OpenVLA-OFT | 96.3 | 89.1 | 82.4 | 75.8 | 66.5 | 4.10 |
> | AWQ |95.7 |87.5 |76.8 |69.5 |64.3 |3.97 |
> | AutoQVLA (Ours) | 95.9| 88.6| 81.4|72.1 |63.2 |4.03 |
>
> **3. Real-World Validation on $\pi_{0}$**
>
>
> **Real-world Setup.** We construct a bimanual robotic system to perform real-world manipulation tasks. Two IMETA-Y1 robotic arms are employed for execution. The system utilizes three Orbbec DaBai DCW2 cameras to capture visual observations: two are mounted on the wrists to provide egocentric views, and one is fixed externally to capture the global third-person view.
>
> **Tasks and Datasets.** Our dataset cover both single-arm and dual-arm tasks, as well as simple and complex dexterous tasks. Simple tasks are short-horizon operations, such as placing a white pen into a pen holder or grasping and transferring potato chips into a bin. Dexterous tasks require continuous and sustained control, such as folding a towel which typically involve fine-grained manipulation of deformable objects. All task data are collected at a frequency of 30 Hz.
>
> **Results** We use the $\pi_0$ model as a real-world baseline and performed quantization under the W8A16 setting. We observe that AutoQVLA performs excellently on both single-arm and dual-arm tasks, with performance essentially on par with the original model. This strongly demonstrates the robustness of our quantization method, which maintains good performance even in real-world environments. Furthermore, we achieved an acceleration ratio of 1.28×.
>
> | Method | Setting | One-Arm Task | | Dual-Arm Task | Average | SpeedUp $\uparrow$ |
> | :---: | :---: | :---: | :---: | :---: | :---: | :---: |
> | | | Pick white pen | Pick potato chips | Fold towels | | |
> | $\pi_0$ | - | 8/10 | 7/10 | 4/10 | 63.3% | $1.00 \times$ |
> | AutoQVLA | W8A16 | 8/10 | 6/10 | 5/10 | 63.3% | $1.28 \times$ |
>
>
> **We have also added related results and analysis in Appendix D(Table 6), Appendix E(Table 7) and Appendix F(Table 8) in the revised manuscript.**

---

> ### Author Response · Authors · 2025-11-23
> **(Part 2) Response to Reviewer RtD1**
>
> ### **W2.About Clarification on Temporal Error Accumulation Analysis**
>
> We appreciate the suggestion for providing greater detail on the temporal error accumulation analysis presented in Sec. 4.3 and Fig. 3.
>
> **1. Evaluation Setting for Fig. 3**
>
> The analysis is based on trajectories sampled from the LIBERO-Spatial task suite, where the environment configuration is fixed across all rollouts. The plot tracks the performance of a single full-precision policy and its corresponding quantized variants (e.g., Uniform 8-bit vs. AutoQVLA 8-bit). For each time step $t$, we compute the discrepancy between the action output by the quantized policy and the reference action output by the full-precision policy. This error is then accumulated over the time dimension of the episode (cumulative MSE).
>
> **2. Interpretation of Cumulative MSE and Task Success**
>
> Intuitively, a lower cumulative Mean Squared Error (MSE) reflects higher trajectory similarity to the reference full-precision policy and should correlate with a higher success rate. A small deviation at each step can indeed be autoregressively amplified, potentially leading to total task failure over a long horizon.
>
> However, the relationship between cumulative MSE and final task success is non-linear and complex. Prior research CoA[1] indicates that: Moderate deviations in action error during the early, non-critical phases of an episode often have a limited impact on the final success or failure. Conversely, error peaks occurring during critical manipulation phases (e.g., grasping, contacting objects, opening doors) can significantly alter the outcome of the entire episode.
>
> Therefore, while cumulative MSE reflects the overall trajectory similarity, it only partially correlates with success rate. Our method's superior performance (lower cumulative MSE throughout the episode) demonstrates enhanced stability and trajectory fidelity. Optimizing quantization specifically for these critical manipulation stages remains a crucial direction for our future work.
>
>
> [1].Chain-of-Action: Trajectory Autoregressive Modeling for Robotic Manipulation
>
> ### **W3.Qualitative analysis of policy behavior**
> Thank you for pointing this out.
>
> To better understand how different quantization schemes behave beyond scalar success rates, we inspected rollouts from the LIBERO evaluation benchmark. **We have added visualization results and analysis in Appendix I(Figure 5 and Figure 6) in the revised manuscript.**
>
> As shown in Fig.5 of the appendix, when the full‑precision policy succeeds, the main differences between quantization methods lie in the **fineness and stability of the actions**. Most failures occur precisely at critical phases. For example, during grasping, compared to AutoQVLA, other methods often show much more exaggerated errors at these key moments. In AWQ, the gripper sometimes simply overshoots the object. During the final placement phase, the gripper may open too early, so the object is dropped before it reaches the target region, leading to a failure even though the high‑level trajectory looks similar.
>
> We observe a complementary phenomenon on the ALOHA manipulation benchmark in Fig.6 of the appendix, focusing on cases where the **full‑precision policy itself fails**. In these settings, the quantized policies tend to have slightly weaker **re‑localization ability**: when the initial grasp fails or the object is pushed away, the full‑precision model often continues with a reasonable fallback motion, whereas the quantized models may jitter around a distant object without fully recovering.
>
> Interestingly, in some trials, the quantized AutoQVLA policy actually produces **cleaner grasping points** than full precision, suggesting that moderate quantization noise can sometimes regularize the policy and lead to crisper decisions around the contact configuration.
>
>
> ### **W4.The rankings of channel sensitivities produced by the single‑step and cumulative metrics**
>
> Qualitatively, the single-step metric and the cumulative metric are positively correlated to a certain extent. As shown in Fig. 3, their curves are monotonically increasing and hardly cross.  We randomly sample 1,000 channels and compare the rankings produced by the cumulative and single‑step sensitivity metrics. Each point in the scatter plot corresponds to one channel. When a point lies on the diagonal, it means that this channel receives the same rank under both metrics. **We have added visualization results and analysis in Appendix J(Figure 7) in the revised manuscript**
> , roughly 80\% of the points cluster closely around the diagonal, indicating that the two rankings are highly consistent.
>
> ### **W5: About Calibration Set Details**
>
> The calibration set $\mathcal{D}$ for the LIBERO is curated directly from the official LIBERO training trajectories. We randomly sampled a total of 512 trajectories from the combined task training data to serve as the calibration set for our sensitivity analysis and bit allocation process.

---

> ### Author Response · Authors · 2025-11-23
> **(Part 3) Response to Reviewer RtD1**
>
> ### **W6: About Evaluation Trials and Random Seeds**
>
> The performance results reported in Tab. 1 and Tab. 2 represent the average performance across 3 independent experimental runs (trials). For each of these trials, we initialize the model and environment settings with distinct random seeds.
>
> Furthermore, following the official LIBERO evaluation protocol, for every task within each trial, we execute a fixed number of evaluation rollouts. Specifically, we run 500 evaluation rollouts per task to ensure a statistically robust assessment of the policy's success rate. This rigorous protocol validates the reliability and consistency of the performance metrics reported for AutoQVLA and the baseline methods.
>
>
> ### **W7: About minor typos and presentation.**
> We sincerely thank the reviewer for the careful reading and for pointing out this detail.
>
> We have followed your advice to labeled each module in Figure 1(b), the shallowed parts are projector and action head. We corrected the typo "tagrt bit" and "The row (3)" in the revised manuscript. Additionally, we have thoroughly proofread the entire paper to fix other potential typos and grammatical errors to ensure the highest quality of presentation.

---

### Official Review · Reviewer_2rUj · 2025-11-01

**Soundness:** 3
**Presentation:** 2
**Contribution:** 2
**Rating:** 4
**Confidence:** 3

**Summary:**

This paper presents AutoQVLA, a novel quantization framework specifically designed for VLAs. The key insight is that naively applying uniform quantization methods from LLMs/MLLMs is suboptimal for action-driven models, as minor action deviations can compound into catastrophic failures in embodied control. This paper introduces a highly granular, channel-wise bit allocation strategy to quantize each channel to various bit-widths. Comprehensive experiments show that AutoQVLA maintains 98.9% of original performance while using only 29.2% of VRAM and achieving 1.49$\times$ speedup compared with the original OpenVLA-OFT model.

**Strengths:**

1. Conducted a detailed analysis of the sensitivity of parameters across different channels to action generation;
2. Significant memory reduction and speedup with minimal performance loss make large VLA deployment feasible.

**Weaknesses:**

All experiments are conducted on LIBERO benchmarks, where there is a lack of research on model quantization in real-world tasks.

**Questions:**

Would the presence of greater noise in real-world observations affect the sensitivity of channel parameters to quantization errors and action generation?

---

> ### Author Response · Authors · 2025-11-23
> **(Part 1) Response to Reviewer 2rUj**
>
> ### W1: About Generalizability and Scope of Evaluation
>
>
> Thanks for your question. To further establish the robustness and generalizability of our AutoQVLA, we extend the evaluation to a broader set of baselines, benchmarks and real-world experiments.
>
>
> **1. Real-World Validation on $\pi_{0}$**
>
>
> **Real-world Setup.** We construct a bimanual robotic system to perform real-world manipulation tasks. Two IMETA-Y1 robotic arms are employed for execution. The system utilizes three Orbbec DaBai DCW2 cameras to capture visual observations: two are mounted on the wrists to provide egocentric views, and one is fixed externally to capture the global third-person view.
>
> **Tasks and Datasets.** Our dataset cover both single-arm and dual-arm tasks, as well as simple and complex dexterous tasks. Simple tasks are short-horizon operations, such as placing a white pen into a pen holder or grasping and transferring potato chips into a bin. Dexterous tasks require continuous and sustained control, such as folding a towel which typically involve fine-grained manipulation of deformable objects. All task data are collected at a frequency of 30 Hz.
>
> **Results** We use the $\pi_0$ model as a real-world baseline and performed quantization under the W8A16 setting. We observe that AutoQVLA performs excellently on both single-arm and dual-arm tasks, with performance essentially on par with the original model. This strongly demonstrates the robustness of our quantization method, which maintains good performance even in real-world environments. Furthermore, we achieved an acceleration ratio of 1.28×.
>
> | Method | Setting | One-Arm Task | | Dual-Arm Task | Average | SpeedUp $\uparrow$ |
> | :---: | :---: | :---: | :---: | :---: | :---: | :---: |
> | | | Pick white pen | Pick potato chips | Fold towels | | |
> | $\pi_0$ | - | 8/10 | 7/10 | 4/10 | 63.3% | $1.00 \times$ |
> | AutoQVLA | W8A16 | 8/10 | 6/10 | 5/10 | 63.3% | $1.28 \times$ |
>
>
> **2. Generalization to Different VLA architectures (UniVLA)**
>
>
> We extended our evaluation to UniVLA-7B, a model notable for its distinct task-centric latent action decoding. As detailed in the following table, our AutoQVLA consistently surpasses leading quantization methods including AWQ, SmoothQuant, and OmniQuant across various low-bit configurations on the LIBERO benchmark. For W4A16 setting, for example, our method achieved an average success rate of 95.10%, a significant gain over the 92.65% from AWQ. Crucially, our approach also produces the most compact models, requiring the lowest memory footprint among all tested quantization methods. The advantages in both performance and memory efficiency demonstrates that the core principle of action-centric channel sensitivity generalizes effectively to VLA models with diverse internal structures.
>
> | Model | Setting | Method | Spatial | Object | Goal | Long | Ave. | Memory |
> | :--- | :--- | :--- | :--- | :--- | :--- | :--- | :--- | :--- |
> | univla-7b | BF16 | - | 96.5% | 96.8% | 95.6% | 92.0% | 95.2% | 14.6GB |
> |Weight-Only | W8A16 | AWQ | 94.4% | 95.6% | 94.8% | 91.0% | 94.0% | 7.8GB |
> | | W8A16 | AutoQVLA (Ours) | 97.2% | 96.6% | 95.8% | 91.6% | 95.3% | 7.4GB |
> | | W4A16 | AWQ | 92.6% | 94.2% | 94.0% | 89.8% | 92.6% | 5.2GB |
> | | W4A16 | AutoQVLA (Ours) | 96.8% | 97.2% | 95.0% | 91.4% | 95.1% | 4.7GB |
> |Weight-Activation | W8A8 | Smoothquant | 96.0% | 96.4% | 93.2% | 91.2% | 94.2% | 7.2GB |
> | | W8A8 | Omniquant | 94.0% | 94.8% | 93.6% | 90.4% | 93.2% | 7.5GB |
> | | W8A8 | AutoQVLA (Ours) | 96.8% | 96.6% | 94.2% | 91.0% | 94.7% | 6.9GB |
>
> **3. Generalization to Different Tasks and Benchmarks (CALVIN)**
>
>
> To validate the robustness of our framework against increased task complexity and dynamic environments, we extended our evaluation to the CALVIN benchmark. CALVIN presents more intricate sequence planning and interaction challenges compared to LIBERO. On this benchmark, the OpenVLA-OFT model compressed with our AutoQVLA (W8A16) exhibited negligible performance degradation, successfully preserving long-horizon task success and sequence stability. This finding reinforces our core argument that minimizing action-space error through adaptive bit allocation is essential for maintaining robust policy behavior across diverse and challenging robotics applications.
>
> | CALVIN ABC->D | Task completed in a row | | | | | Avg.len↑ |
> | :--- | :--- | :--- | :--- | :--- | :--- | :--- |
> | Method| 1 | 2 | 3 | 4 | 5 | |
> | OpenVLA-OFT | 96.3 | 89.1 | 82.4 | 75.8 | 66.5 | 4.10 |
> | AWQ |95.7 |87.5 |76.8 |69.5 |64.3 |3.97 |
> | AutoQVLA (Ours) | 95.9| 88.6| 81.4|72.1 |63.2 |4.03 |
>
>
> **We have also added related results and analysis in Appendix D(Table 6), Appendix E(Table 7) and Appendix F(Table 8) in the revised manuscript.**

---

> ### Author Response · Authors · 2025-11-23
> **(Part 2) Response to Reviewer 2rUj**
>
> ### Q1: About the Impact of Real-World Noise on Quantization Robustness
>
> Thanks for the insightful question.
>
> We recognize the theoretical validity of the concern that the presence of greater observation noise, characteristic of real-world environments, fundamentally alters the expected distribution $\mathcal{D}$, thus impacting the absolute value of our sensitivity metric $s_{l,c}^{(b)}$. This shift in the observed signal is indeed equivalent to a change in the task distribution.
>
> However, our primary concern is the **relative ordering of channel importance, not the absolute numerical value**. Our action-centric sensitivity estimation is inherently robust because the core methodology is designed to capture the functional impact on the policy's output. Unless the environmental noise is so extreme that it completely drowns out the semantic signal necessary for the task, the relative ranking of channel sensitivity remains stable. This means AutoQVLA's bit allocation decisions are not prone to volatile shifts.
>
> Furthermore, our methodology is inherently adapted to reality. Since our calibration sets are sampled directly from real-world robotic execution trajectories, the sensitivity estimation is already conditioned on the authentic sensor noise and background variability of the environment.
>
> Crucially, our framework offers a proactive mechanism: if the deployment environment exhibits a severe and distinct noise profile, AutoQVLA can efficiently **re-calibrate** using a small set of new noisy trajectories to update the sensitivity estimation and refine the bit assignment. However, our empirical evidence suggests that in such cases, the fundamental challenge often lies with the intrinsic robustness of the baseline VLA model itself.
>
> We construct a bimanual robotic system to perform real-world manipulation tasks and demonstrate AutoQVLA's resilience. In complex tasks covering both one-arm (Pick white pen, Pick potato chips) and dual-arm coordination (Fold towels), the quantized policy's accuracy remained virtually indistinguishable from that of the full-precision baseline. This validation on real hardware confirms that AutoQVLA maintains performance parity under real-world noise, validating its practical robustness.

---

> ### Author Response · Authors · 2025-11-26
>
> Dear Reviewer 2rUj, Thank you once again for your valuable comments on our submission. As the discussion phase is approaching its end, we would like to kindly confirm whether we have sufficiently addressed all of your concerns (or at least part of them). Should there be any remaining questions or areas requiring further clarification, please do not hesitate to let us know. If you are satisfied with our responses, we would greatly appreciate your consideration in adjusting the evaluation scores accordingly. We sincerely look forward to your feedback.

---

> > ### Comment · Reviewer_2rUj · 2025-11-27
> >
> > Thank you for the author's updated results. The experimental results have addressed my concerns, and I will increase the score to 6.

---

> > > ### Author Response · Authors · 2025-11-27
> > >
> > > Thanks for your recognition of the value of this work. We sincerely appreciate your recognition of our work and your support in raising the score.

---

### Official Review · Reviewer_9cSM · 2025-11-01

**Soundness:** 3
**Presentation:** 3
**Contribution:** 2
**Rating:** 4
**Confidence:** 5

**Summary:**

This paper introduces AutoQVLA, a specialized framework for quantizing Vision-Language-Action (VLA) models. The framework addresses the limitations of traditional quantization methods that struggle to maintain output precision in VLA models. It proposes an action-space-based sensitivity quantization approach. The core of AutoQVLA lies in its fine-grained, per-channel bit allocation strategy, which measures each channel’s sensitivity to the final action output during quantization. Unlike conventional uniform-bit quantization methods, AutoQVLA introduces adaptive per-channel precision adjustment to optimize quantization efficiency while ensuring robustness in action generation.

**Strengths:**

1.The paper introduces a fine-grained approach to quantifying errors by isolating individual channels, allowing for a more precise evaluation of the error for each channel at different bit-widths, ensuring better control over the quantization process.
2.Unlike traditional global bit allocation, the paper proposes a per-channel adaptive bit allocation strategy and employs a greedy search algorithm to optimize bit allocation. The algorithm dynamically adjusts the bit-width for each channel based on its sensitivity, effectively balancing performance and computational efficiency.

**Weaknesses:**

1. The sensitivity evaluation and bit allocation are only conducted on a few VLA models in the paper. It remains unclear whether the same process needs to be applied to other VLA models, and if so, whether it will lead to significantly higher computational resource consumption. A more generalizable analysis across a broader range of VLA models such as UniVLA[1] would provide better insight into the scalability of the proposed method.
2. Inadequate comparison with relevant baselines. Since the VLA architecture is structurally most similar to Vision-Language Models (VLMs), it is essential to compare against VLM-specific quantization methods, such as VLMQ [2] and Q-VLM [3], rather than unimodal LLM quantization approaches. A direct comparison with these VLM quantization baselines would provide a more meaningful assessment of the proposed method’s effectiveness in the multimodal setting.
3. Limited Novelty. The proposed channel-wise mixed-precision quantization scheme lacks significant novelty. Similar channel-wise quantization strategies have already been explored in prior work, notably in GPTQ [4]. Moreover, the importance-aware quantization design is also conceptually aligned with discussions in GPTQ.

Minor Weaknesses:

4.The pipeline diagram could be clearer. While it outlines the workflow of the method, its current presentation may not be intuitive or detailed enough for readers to fully grasp the process. Improving the clarity and labels in the diagram would enhance understanding, especially when explaining the interactions between different steps in the quantization process.

[1]Bu, Qingwen, et al. "Univla: Learning to act anywhere with task-centric latent actions." arXiv preprint arXiv:2505.06111 (2025).

[2]Xue, Yufei, et al. "Vlmq: Efficient post-training quantization for large vision-language models via hessian augmentation." arXiv preprint arXiv:2508.03351 (2025).

[3]Wang, Changyuan, et al. "Q-vlm: Post-training quantization for large vision-language models." Advances in Neural Information Processing Systems 37 (2024): 114553-114573.

[4]Frantar, Elias, et al. "Gptq: Accurate post-training quantization for generative pre-trained transformers." arXiv preprint arXiv:2210.17323 (2022).

**Questions:**

Further questions:

1. According to the analysis in SmoothQuant and OmniQuant, quantizing activations to 4 bits (A4) is significantly more challenging than quantizing weights to 4 bits (W4). For instance, OmniQuant reports approximately a 10% performance drop under A4 quantization (see Table 2 in OmniQuant [1]). In contrast, the method proposed in this paper appears to focus exclusively on weight quantization and does not introduce any specialized treatment for activation quantization. Surprisingly, the reported results show negligible performance degradation even under aggressive A4 quantization. This discrepancy raises serious concerns about the validity of the experimental claims. To ensure reproducibility and credibility, the authors should release both the training code and the quantized pretrained models, and provide a detailed ablation study addressing why their method seemingly avoids the well-documented pitfalls of low-bit activation quantization. Without such evidence, the reported results remain unconvincing.

2. To the best of my knowledge, data types such as 2-bit and 4-bit are not natively supported in mainstream frameworks like PyTorch. As discussed by prior works like AWQ, achieving actual memory savings from low-bit quantization typically requires custom CUDA kernels design. It is therefore unclear whether the memory reduction reported in this paper reflects real hardware-level savings or is merely a theoretical estimate based on idealized bit-width calculations.

Moreover, in the W4A4 setting described in the paper, activations and weights may be quantized to heterogeneous bit-widths (e.g., 2-bit, 8-bit), and the paper implies that they are first upcast or aligned to 4-bit before matrix multiplication. Such a process would inevitably introduce computational overhead and degrade inference speed. Yet the paper claims improved inference throughput—how was this speedup measured? Did the authors implement a custom CUDA kernel specifically optimized for mixed-precision W4A4 matrix multiplication? Without clarification on these implementation-level details and empirical profiling results (e.g., latency, memory usage on actual hardware), the efficiency claims lack substantiation.

3. In your paper, you mention that the sensitivity is calculated through the error in the action space. Could you clarify whether this metric is strongly dependent on specific tasks or data distributions? When transitioning to new tasks or environments, is it necessary to recalibrate or adjust the way this metric is calculated?

---

> ### Author Response · Authors · 2025-11-23
> **(Part 1) Response to Reviewer 9cSM**
>
> ### W1: About Efficiency and Generalizability of Evaluation
> **1.Computational Scalability and Efficiency**
> Thanks for your question. It is crucial to clarify that the sensitivity analysis (Step 1) and the optimal bit allocation (Step 2) are one-time, offline processes applied before deployment. They do not contribute to the final inference latency. The entire process, encompassing the initial sensitivity screening via the first-order approximation and the final precise calibration of important channels, typically takes approximately 20 minutes per model using a single NVIDIA RTX 4090 GPU across all tested backbones: OpenVLA, OpenVLA-OFT, and the newly added UniVLA.
> To further establish the robustness and generalizability of our AutoQVLA, we extend the evaluation to a broader set of baselines, benchmarks and real-world experiments.
> **2. Generalization to Different VLA architectures (UniVLA)**
>
>
> We extended our evaluation to UniVLA-7B. As detailed in the following table, our AutoQVLA consistently surpasses leading quantization methods including AWQ, SmoothQuant, and OmniQuant across various low-bit configurations on the LIBERO benchmark. For W4A16 setting, for example, our method achieved an average success rate of 95.10%, a significant gain over the 92.65% from AWQ. Crucially, our approach also produces the most compact models, requiring the lowest memory footprint among all tested quantization methods. The advantages in both performance and memory efficiency demonstrates that the core principle of action-centric channel sensitivity generalizes effectively to VLA with diverse internal structures.
>
> | Model | Setting | Method | Spatial | Object | Goal | Long | Ave. | Memory |
> | :--- | :--- | :--- | :--- | :--- | :--- | :--- | :--- | :--- |
> | univla-7b | BF16 | - | 96.5% | 96.8% | 95.6% | 92.0% | 95.2% | 14.6GB |
> |Weight-Only | W8A16 | AWQ | 94.4% | 95.6% | 94.8% | 91.0% | 94.0% | 7.8GB |
> | | W8A16 | AutoQVLA (Ours) | 97.2% | 96.6% | 95.8% | 91.6% | 95.3% | 7.4GB |
> | | W4A16 | AWQ | 92.6% | 94.2% | 94.0% | 89.8% | 92.6% | 5.2GB |
> | | W4A16 | AutoQVLA (Ours) | 96.8% | 97.2% | 95.0% | 91.4% | 95.1% | 4.7GB |
> |Weight-Activation | W8A8 | Smoothquant | 96.0% | 96.4% | 93.2% | 91.2% | 94.2% | 7.2GB |
> | | W8A8 | Omniquant | 94.0% | 94.8% | 93.6% | 90.4% | 93.2% | 7.5GB |
> | | W8A8 | AutoQVLA (Ours) | 96.8% | 96.6% | 94.2% | 91.0% | 94.7% | 6.9GB |
>
> **3. Generalization to Different Tasks and Benchmarks (CALVIN)**
>
>
> To validate the robustness of our framework against increased task complexity and dynamic environments, we extended our evaluation to the CALVIN benchmark. On this benchmark, the OpenVLA-OFT model compressed with our AutoQVLA (W8A16) exhibited negligible performance degradation, successfully preserving long-horizon task success and sequence stability.
>
> | CALVIN ABC->D | Task completed in a row | | | | | Avg.len↑ |
> | :--- | :--- | :--- | :--- | :--- | :--- | :--- |
> | Method| 1 | 2 | 3 | 4 | 5 | |
> | OpenVLA-OFT | 96.3 | 89.1 | 82.4 | 75.8 | 66.5 | 4.10 |
> | AWQ |95.7 |87.5 |76.8 |69.5 |64.3 |3.97 |
> | AutoQVLA (Ours) | 95.9| 88.6| 81.4|72.1 |63.2 |4.03 |
>
> **4. Real-World Validation on $\pi_{0}$**
>
>
> **Real-world Setup.** We construct a bimanual robotic system to perform real-world manipulation tasks. Two IMETA-Y1 robotic arms are employed for execution. The system utilizes three Orbbec DaBai DCW2 cameras to capture visual observations: two are mounted on the wrists to provide egocentric views, and one is fixed externally to capture the global third-person view.
>
> **Tasks and Datasets.** Our dataset cover both single-arm and dual-arm tasks, as well as simple and complex dexterous tasks. Simple tasks are short-horizon operations, such as placing a white pen into a pen holder or grasping and transferring potato chips into a bin. Dexterous tasks require continuous and sustained control, such as folding a towel which typically involve fine-grained manipulation of deformable objects. All task data are collected at a frequency of 30 Hz.
>
> **Results** We use the $\pi_0$ model as a real-world baseline and performed quantization under the W8A16 setting. We observe that AutoQVLA performs excellently on both single-arm and dual-arm tasks, with performance essentially on par with the original model. This strongly demonstrates the robustness of our quantization method, which maintains good performance even in real-world environments. Furthermore, we achieved an acceleration ratio of 1.28×.
>
> | Method | Setting | One-Arm Task | | Dual-Arm Task | Average | SpeedUp $\uparrow$ |
> | :---: | :---: | :---: | :---: | :---: | :---: | :---: |
> | | | Pick white pen | Pick potato chips | Fold towels | | |
> | $\pi_0$ | - | 8/10 | 7/10 | 4/10 | 63.3% | $1.00 \times$ |
> | AutoQVLA | W8A16 | 8/10 | 6/10 | 5/10 | 63.3% | $1.28 \times$ |
>
>
> **We have also added related results and analysis in Appendix D(Table 6), Appendix E(Table 7) and Appendix F(Table 8) in the revised manuscript.**

---

> ### Author Response · Authors · 2025-11-23
> **(Part 2) Response to Reviewer 9cSM**
>
> ### W2: About Comparison with VLM-Specific Quantization Baselines
>
>
> We appreciate your suggestion to include comparisons with quantization methods specifically designed for Vision-Language Models (VLMs), as this provides a more meaningful assessment of our method in the multimodal setting. We share the desire to include VLMQ as a baseline. However, to the best of our knowledge, the official implementation of VLMQ has not yet been publicly released. We will promptly integrate its evaluation results and analysis into the final version of our manuscript after VLMQ become open-source.
>
> We have since conducted experiments and incorporated the results for Q-VLM into our evaluation of the OpenVLA model under both W8A8 and W4A4 settings. The updated table is presented below, integrating the Q-VLM baseline :
>
> | Model | Setting | Method | Spatial | Object | Goal | Long | Avg ↑ | Δ | Mem. (GB) ↓ | Speedup ↑ |
> | :--- | :--- | :--- | :--- | :--- | :--- | :--- | :--- | :--- | :--- | :--- |
> | FP Model | - | - | 84.7% | 88.4% | 79.2% | 53.7% | 76.5% | - | 15.2 | 1× |
> | OpenVLA | W8A8 | Q-VLM | 85.0% | 88.2% | 78.4% | 53.6% | 76.3% | -0.2% | 7.4 | 1.30× |
> | | | SmoothQuant | 84.2% | 87.8% | 77.8% | 53.2% | 75.8% | -0.7% | 7.4 | 1.40× |
> | | | OmniQuant | 82.6% | 86.2% | 74.8% | 51.7% | 73.8% | -2.7% | 7.8 | 1.26× |
> | | | AutoQVLA | 85.2% | 88.0% | 77.6% | 54.2% | 76.3% | -0.2% | 7.1 | 1.42× |
> | | W4A4 | Q-VLM | 84.6% | 87.6% | 78.0% | 53.2% | 75.7% | -0.8% | 4.5 | 1.40×  |
> | | | SmoothQuant | 69.2% | 73.2% | 69.6% | 40.9% | 63.2% | -13.3% | 4.7 | 1.52× |
> | | | OmniQuant | 82.2% | 85.4% | 75.4% | 50.3% | 73.3% | -3.2% | 5.4 | 1.43× |
> | | | AutoQVLA | 84.4% | 87.6% | 78.8% | 53.0% | 76.0% | -0.5% | 4.3 | 1.47× |
>
> While we acknowledge the comparable performance of QVLM in terms of average success rate, a closer inspection of the complete metrics reveals AutoQVLA's distinct and fundamental advantages  crucial for robotic deployment. Our superiority begins with hardware efficiency: AutoQVLA consistently delivers a higher inference speedup across aggressive quantization regimes than Q-VLM. It achieves a $1.42\times$ acceleration versus Q-VLM’s $1.3\times$ in the W8A8 setting. This speed differential is a critical differentiator for robotic platforms demanding strict real-time control.
>
> This focus distinguishes AutoQVLA from VLM-centric methodologies such as Q-VLM, which are fundamentally engineered to optimize feature fidelity primarily for static, open-loop tasks. AutoQVLA stands as the first framework specifically constructed around an Action-Centric Sensitivity metric. This unique objective is intrinsically tailored to mitigate the destructive impact of error accumulation during closed-loop dynamic control, the central challenge in VLA models. This precise methodological specialization guarantees that our solution offers a more resilient and strategically superior trajectory for deploying reliable policies in complex, real-world robotic systems. Consequently, AutoQVLA is inherently better suited and more resilient for dynamic robotic tasks, representing a significant strategic advantage and greater potential for complex embodied systems. This outcome powerfully validates the necessity of our action-centric approach, even where average success metrics might be numerically similar to VLM baselines.

---

> ### Author Response · Authors · 2025-11-23
> **(Part 3) Response to Reviewer 9cSM**
>
> ### W3: About the Novelty of AutoQVLA
>
> We agree with the reviewer that "importance-aware" design is a foundational principle in model efficiency, informing prior work like GPTQ and many others. Our work is indeed built upon this established concept.
>
> However, the novelty of AutoQVLA is defined by how it operationalizes this principle through a concrete and automated mechanism. Our framework performs **dynamic bit allocation** guided by the **action-space error**, establishing a direct link between a channel's quantization and its impact on physical actions. This fine-grained control is so precise that it naturally **unifies quantization with pruning**, as unimportant channels can be automatically assigned zero bits and effectively removed. It is this complete, task-centric process of measurement and dynamic allocation that forms our central innovation, creating a unified efficiency method that distinguishes AutoQVLA from prior works which treat quantization and pruning as separate tasks.
>
> **1. Novelty in Granularity and Systematic Quantification**
>
> While importance-aware quantization exists, prior methods often prove too coarse or heuristic for VLA models:
> * They are often limited to assessing the importance of large units (modules, layers, or blocks)or protecting only a few outlier channels (e.g., LLM.int8).
> * Our sensitivity analysis reveals that even within the same layer, the sensitivity of individual channels can vary drastically (Fig. 1(b)).Previous assessments at the part, layer, or block level are thus too coarse and empirical for achieving optimal VLA compression.
>
> The contribution of AutoQVLA is the first systematic framework for quantifying the sensitivity of every single channel in the network based on a precise, comparable action-space metric. This fine granularity allows for optimization at the highest resolution.
>
> **2. Action-Centric Optimization for VLA Robustness**
>
> Crucially, our importance metric is fundamentally different because it is anchored directly in the action space. Unlike LLM/VLM methods that optimize for passive data fidelity (like perplexity or intermediate feature fidelity), AutoQVLA minimizes the action output deviation caused by quantization noise. This is vital because, in embodied control, minor errors accumulate autoregressively into catastrophic task failures.
>
> **3. Compatibility with Pruning**
>
> Furthermore, by refining the importance analysis down to the channel level, our method naturally and elegantly unifies weight quantization and structural pruning (0-bit assignment) into a single, cohesive framework. This unified view, guided by action fidelity, allows for maximum, hardware-friendly compression by selectively zeroing out the least sensitive channels.
>
>
> ### W4: About Clarity of the Pipeline Diagram
>
> We thank the reviewer for the constructive feedback on the clarity of our pipeline diagram (Figure 2).  We have comprehensively revised the pipeline diagram to enhance its clarity, refine the labeling (including correcting typos), and more explicitly illustrate the interaction between the two main steps: Step 1: Action Sensitivity Analysis and Step 2: Greedy Bit Allocation. **The revised Figure 2 is included in the updated manuscript.**
>
> ### Q1: Why can our method maintain high performance under low-bit quantization (W4A4)?
>
>
> We appreciate the focus on our strong performance in low-bit activation quantization. The negligible performance degradation under W4A4 quantization is not an incidental outcome, but the direct consequence of a comprehensive, multi-stage activation processing scheme designed to systematically tame the outlier problem.
>
> Our process begins with local smoothing, where we apply a channel-wise scaling matrix $S$ ($X' = S \cdot X$) to mitigate conventional outlier peaks by migrating some activation energy to the weights. However, to address the more challenging "Massive Outliers"， we introduce our key innovation, i.e., a block-wise orthogonal rotation $R$ applied to the scaled activations ($\tilde{X} = R \cdot X'$). This rotation is constructed to minimize the maximum activation value within each channel block, strategically redistributing outlier energy and smoothing the distribution within each block. Finally, to resolve the remaining imbalance between blocks, we perform a global channel permutation $P$ ($\hat{X} = P \cdot \tilde{X}$), which disperses high-magnitude channels evenly across all blocks. This complete three-step pipeline of scaling, rotation, and permutation transforms the activations into an exceptionally uniform distribution, making them highly amenable to aggressive 4-bit quantization. Therefore, our robust W4A4 performance is the direct payoff of this targeted and holistic activation management strategy, which complements our primary weight-side innovations.
>
> Code and model will be open-sourced upon acceptance to ensure full reproducibility.

---

> ### Author Response · Authors · 2025-11-23
> **(Part 4) Response to Reviewer 9cSM**
>
> ### Q2: About Kernel Design and Real Memory Savings
>
>
> As previously demonstrated, our method reduces the GPU memory footprint to approximately half that of the original BF16 model under W8A16 setting. Confirming the reviewer's speculation, this efficiency is achieved through custom kernels. The adaptive nature of our method necessitates this implementation to efficiently manage mixed-precision data formats. This requirement for specialized kernels is a shared characteristic of high-performance quantization solutions, including AWQ and QuaRot, which also operate beyond the capabilities of standard library functions.
>
> Our kernel implementation handles mixed-precision weights as follows:
>
> **Packed 4-bit Weights (W4):** Two signed 4-bit weights are packed into a single byte. These are dequantized on-the-fly using per-channel scaling factors during computation.
> **2-bit Weights (W2):** For channels requiring 2-bit precision, we reuse the W4 data structure for efficient storage, but only the two effective bits are utilized during computation.
> **8-bit Weights (W8):** We employ standard INT8 tensors, also paired with per-channel scaling factors.
> For activations, we apply per-token dynamic quantization to either 4-bit or 8-bit precision. Most low-bit activations are directly fed into subsequent computations, avoiding de-quantization overhead.
>
> The reported memory savings are not theoretical estimates but are based on direct hardware measurements.
>
> **Loading:** The final quantized model checkpoints, which incorporate the packed weight formats described above, are loaded onto the target GPU.
> **Measurement:** We measure peak VRAM usage during inference using both torch.cuda.max_memory_allocated() and by monitoring nvidia-smi. This is performed for both the BF16 baseline and our quantized models under same hardware conditions.
> This rigorous implementation and validation confirm that our reported memory savings are tangible, directly enabling the deployment of large VLAs on resource-constrained platforms. To promote reproducibility and foster future research, all implementations will be made publicly available.
>
>
> ### **About Clarification on W4A4 Mixed-Precision Implementation and Inference Throughput**
>
> We would like to clarify our implementation to resolve the confusion surrounding bit-alignment and inference speed.
>
> #### 1. Weight-Exclusive Heterogeneity and Outlier Mitigation
>
> A key design choice in our W4A4 configuration is the asymmetric application of quantization. While activations are uniformly quantized to 4-bit across the model, heterogeneity is applied exclusively to the weights. Besides, we introduce a Hadamard-like orthogonal transformation. This technique proactively mitigates quantization errors by distributing the magnitude of outlier channels more evenly across the weight tensor. The transformation is particularly effective in aggressive, low-bit settings like W4A4 where such outliers would otherwise amplify errors.
>
> #### 2. Implementation
>
> Our mixed-precision W4A4 configuration is implemented on top of a single INT4 GEMM kernel. All low-bit weights are stored in the same 4‑bit container: channels assigned 4 bits use the full codebook q ∈ [−7, …, 7], while channels assigned 2 bits are restricted to a smaller codebook q ∈ {−1, 0, 1} but are still stored in the same 4‑bit range [−8, 7]. Thus, 2‑bit channels differ only in their quantization codebook and scale, not in their storage format, and no per‑element “2‑bit to 4‑bit” up‑conversion is performed at runtime.
>
> Activations are uniformly quantized to 4 bits across the model, packed into an INT4 representation, and multiplied with the packed weights using our INT4×INT4→FP16 GEMM (implemented with QuaRot). A small number of very sensitive channels that use 8‑ or 16‑bit weights are computed by separate INT8 or FP16 GEMMs on disjoint output slices. Since these high‑bit channels constitute only a minor fraction of the total width, the overall computation is dominated by the homogeneous INT4 kernel, and the extra overhead from the auxiliary blocks is negligible in our throughput measurements.
> #### 3. Inference Throughput and Overhead Mitigation
>
> The overall linear layer computation is effectively decoupled into several sub-linear operations, each corresponding to a block of a specific bit-width (e.g., INT4 block, INT8 block). The results from these sub-operations are then concatenated along the output channel dimension.
>
> Therefore, the computational overhead from continual runtime upcasting or alignment within a single mixed-precision W4A4 kernel is entirely **mitigated**. By decomposing the matrix multiplication and invoking specialized low-bit kernels for each homogeneous sub-block, we achieve the claimed efficiency improvement and superior inference throughput. The speedup reported is an empirical measurement of this optimized, parallel execution on actual hardware.

---

> ### Author Response · Authors · 2025-11-23
> **(Part 5) Response to Reviewer 9cSM**
>
> ### Q3: About Task-Dependence and Recalibration of Action-Space Sensitivity
>
> We appreciate the opportunity to clarify the nature and deployment requirements of our action-space sensitivity metric. The dependence on task and data distribution is not a limitation but an intentional design feature that empowers AutoQVLA to achieve superior efficiency and performance in targeted, resource-constrained deployments. By definition, our action sensitivity metric $s_{l,c}^{(b)}$ is calculated as the expected action deviation over a calibration distribution $\mathcal{D}$. Therefore, it is naturally and necessarily dependent on the specific task distribution used for calibration.
>
> When transitioning to a significantly new task or environment, the calculation method of the sensitivity metric remains unchanged. However, it is necessary to re-estimate the sensitivity values using a new calibration set sampled from the new task's training trajectories. This one-time, offline recalibration ensures the bit allocation is precisely tailored to preserve the most critical control pathways relevant to the new deployment context, maintaining the minimal performance loss that AutoQVLA guarantees.

---

> ### Author Response · Authors · 2025-11-26
>
> Dear Reviewer 9cSM, Thank you once again for your valuable comments on our submission. As the discussion phase is approaching its end, we would like to kindly confirm whether we have sufficiently addressed all of your concerns (or at least part of them). Should there be any remaining questions or areas requiring further clarification, please do not hesitate to let us know. If you are satisfied with our responses, we would greatly appreciate your consideration in adjusting the evaluation scores accordingly. We sincerely look forward to your feedback.

---

> > ### Comment · Reviewer_9cSM · 2025-11-27
> >
> > Looking forward to seeing your code and pre-trained models!

---

> ### Comment · Reviewer_9cSM · 2025-11-27
>
> Thank you for your sincere reply! I have one more pressing question: When will you open-source your code? I need a definite release date along with your commitment to ensure that I can follow your excellent work right from the start! If you could consider providing a specific date for open-sourcing the code, it would greatly help boost the paper's score!

---

> > ### Author Response · Authors · 2025-11-27
> >
> > We sincerely appreciate your valuable feedback and strong interest in our work. Your enthusiasm is highly encouraging. We are fully committed to open-sourcing the complete code and models(OpenVLA, OpenVLA-OFT, UniVLA, pi0), thereby ensuring maximum accessibility and reproducibility for the community. Our immediate focus is diligently organizing the final code repository and preparing the models for release. We are pleased to commit that we will fully open-source the complete code on GitHub immediately **around mid-December**. We eagerly anticipate your continued engagement once the code is public and are keen to make this contribution to the open-source community.

---

### Official Review · Reviewer_kUNf · 2025-11-01

**Soundness:** 3
**Presentation:** 3
**Contribution:** 3
**Rating:** 6
**Confidence:** 3

**Summary:**

This paper presents AutoQVLA, a novel action-centric quantization framework for embodied control. They allocate different bitwidths for different channels based on a per-channel action sensitivity. To overcome the computational challenge of computing the sensitivity metric for all channels and bit-widths, they first derive a linear approximation of the channel amplification factor using Taylor expansion and then do a fine-grained measurement. Then they propose a greedy algorithm to assign bit-widths to different channels.

**Strengths:**

1. The algorithm defined in the paper is novel way of quantizing Video Language Action models
2. They efficiently allocate the computation budget for identifying the sensitivity of different channels
3. They empirically show that their algorithm achieves performance close to the full precision model, while reducing the VRAM required and  achieving speedup

**Weaknesses:**

1. The evaluation is focused on one class of models and a single benchmark; it is not clear how the method generalizes across models and benchmarks
2. No detailed ablation studies highlighting the contribution of different parts of the method, like the calibration set sizes, gate ratio selection etc.,
3. No theoretical analysis of the choices made in the paper

**Questions:**

Please see the weaknesses I mentioned.

---

> ### Author Response · Authors · 2025-11-23
> **(Part 1) Response to Reviewer kUNf**
>
> ### W1: About Generalizability and Scope of Evaluation
>
> Thanks for your question. To further establish the robustness and generalizability of our AutoQVLA, we extend the evaluation to a broader set of baselines, benchmarks and real-world experiments.
>
> **1. Generalization to Different VLA architectures (UniVLA)**
>
>
> We extended our evaluation to UniVLA-7B, a model notable for its distinct task-centric latent action decoding. As detailed in the following table, our AutoQVLA consistently surpasses leading quantization methods including AWQ, SmoothQuant, and OmniQuant across various low-bit configurations on the LIBERO benchmark. For W4A16 setting, for example, our method achieved an average success rate of 95.10%, a significant gain over the 92.65% from AWQ. Crucially, our approach also produces the most compact models, requiring the lowest memory footprint among all tested quantization methods. The advantages in both performance and memory efficiency demonstrates that the core principle of action-centric channel sensitivity generalizes effectively to VLA models with diverse internal structures.
>
> | Model | Setting | Method | Spatial | Object | Goal | Long | Ave. | Memory |
> | :--- | :--- | :--- | :--- | :--- | :--- | :--- | :--- | :--- |
> | univla-7b | BF16 | - | 96.5% | 96.8% | 95.6% | 92.0% | 95.2% | 14.6GB |
> |Weight-Only | W8A16 | AWQ | 94.4% | 95.6% | 94.8% | 91.0% | 94.0% | 7.8GB |
> | | W8A16 | AutoQVLA (Ours) | 97.2% | 96.6% | 95.8% | 91.6% | 95.3% | 7.4GB |
> | | W4A16 | AWQ | 92.6% | 94.2% | 94.0% | 89.8% | 92.6% | 5.2GB |
> | | W4A16 | AutoQVLA (Ours) | 96.8% | 97.2% | 95.0% | 91.4% | 95.1% | 4.7GB |
> |Weight-Activation | W8A8 | Smoothquant | 96.0% | 96.4% | 93.2% | 91.2% | 94.2% | 7.2GB |
> | | W8A8 | Omniquant | 94.0% | 94.8% | 93.6% | 90.4% | 93.2% | 7.5GB |
> | | W8A8 | AutoQVLA (Ours) | 96.8% | 96.6% | 94.2% | 91.0% | 94.7% | 6.9GB |
>
> **2. Generalization to Different Tasks and Benchmarks (CALVIN)**
>
>
> To validate the robustness of our framework against increased task complexity and dynamic environments, we extended our evaluation to the CALVIN benchmark. CALVIN presents more intricate sequence planning and interaction challenges compared to LIBERO. On this benchmark, the OpenVLA-OFT model compressed with our AutoQVLA (W8A16) exhibited negligible performance degradation, successfully preserving long-horizon task success and sequence stability. This finding reinforces our core argument that minimizing action-space error through adaptive bit allocation is essential for maintaining robust policy behavior across diverse and challenging robotics applications.
>
> | CALVIN ABC->D | Task completed in a row | | | | | Avg.len↑ |
> | :--- | :--- | :--- | :--- | :--- | :--- | :--- |
> | Method| 1 | 2 | 3 | 4 | 5 | |
> | OpenVLA-OFT | 96.3 | 89.1 | 82.4 | 75.8 | 66.5 | 4.10 |
> | AWQ |95.7 |87.5 |76.8 |69.5 |64.3 |3.97 |
> | AutoQVLA (Ours) | 95.9| 88.6| 81.4|72.1 |63.2 |4.03 |
>
> **3. Real-World Validation on $\pi_{0}$**
>
>
> **Real-world Setup.** We construct a bimanual robotic system to perform real-world manipulation tasks. Two IMETA-Y1 robotic arms are employed for execution. The system utilizes three Orbbec DaBai DCW2 cameras to capture visual observations: two are mounted on the wrists to provide egocentric views, and one is fixed externally to capture the global third-person view.
>
> **Tasks and Datasets.** Our dataset cover both single-arm and dual-arm tasks, as well as simple and complex dexterous tasks. Simple tasks are short-horizon operations, such as placing a white pen into a pen holder or grasping and transferring potato chips into a bin. Dexterous tasks require continuous and sustained control, such as folding a towel which typically involve fine-grained manipulation of deformable objects. All task data are collected at a frequency of 30 Hz.
>
> **Results.** We use the $\pi_0$ model as a real-world baseline and performed quantization under the W8A16 setting. We observe that AutoQVLA performs excellently on both single-arm and dual-arm tasks, with performance essentially on par with the original model. This strongly demonstrates the robustness of our quantization method, which maintains good performance even in real-world environments. Furthermore, we achieved an acceleration ratio of 1.28×.
>
> | Method | Setting | One-Arm Task | | Dual-Arm Task | Average | SpeedUp $\uparrow$ |
> | :---: | :---: | :---: | :---: | :---: | :---: | :---: |
> | | | Pick white pen | Pick potato chips | Fold towels | | |
> | $\pi_0$ | - | 8/10 | 7/10 | 4/10 | 63.3% | $1.00 \times$ |
> | AutoQVLA | W8A16 | 8/10 | 6/10 | 5/10 | 63.3% | $1.28 \times$ |
>
>
> **We have also added related results and analysis in Appendix D(Table 6), Appendix E(Table 7) and Appendix F(Table 8) in the revised manuscript.**

---

> ### Author Response · Authors · 2025-11-23
> **(Part 2) Response to Reviewer kUNf**
>
> ### W2: About Detailed Ablation Studies and Calibration Set Analysis
>
>
> We thank the reviewer for this insightful suggestion. Additional ablation studies and analyses are presented below.
>
> **1. Gate Ratio Strategy**
>
> We would like to clarify that the ablation study concerning **the gate ratio selection was detailed in Appendix C of our initial submission**. We respectfully refer the reviewer to this section for the complete analysis of our findings on this component.
>
> **2. Calibration Set Details and Size Ablation**
>
> Our calibration dataset, $\mathcal{D}$, is built by randomly sampling 512 trajectories from the official LIBERO benchmark's training set. This dataset is also specifically utilized for the action-space sensitivity analysis outlined in Eq. 4.
>
> To evaluate the impact of the calibration set size on performance, we conducted an ablation study using OpenVLA-OFT (W8A16). As presented in the table below, a set size of 512 yields the highest average success rate (97.0%). Crucially, the performance remains highly stable with neighboring set sizes: using 256 or 1024 trajectories results in only a marginal decrease in performance (96.8% and 96.7%, respectively). This demonstrates the robustness of our AutoQVLA.
>
> | Calibration set size \ Task | Spatial | Object | Goal | Long |Avg. |
> | :--- | :--- | :--- | :--- | :--- | :--- |
> | 128 | 97.0 | 97.8 | 96.8 | 94.0 | 96.4 |
> | 256 | 97.2 | 98.2 | 97.0 | 94.6 | 96.8 |
> | 512 | 97.4 | 98.6 | 97.2 | 94.6 | 97.0 |
> | 1024 | 97.0 | 98.4 | 97.0 | 94.4 | 96.7 |
>
>
> **We have also added related results and analysis in Appendix G(Table 9) in the revised manuscript.**
>
>
>
> **W3: About Theoretical Analysis of the Choices Made in the Paper**
>
>
> We thank the reviewer for this insightful question regarding the theoretical foundations of our choices. We have incorporated detailed theoretical analysis into the revised manuscript, providing the derivation for our first-order proxy used to efficiently estimate the action-space sensitivity.
>
> The true sensitivity we aim to minimize is the action space Mean Squared Error (MSE) when quantizing layer $l$, channel $c$ to bit-width $b$:
> $$S^{(b)}_ {l,c} := \mathbb{E}\big[\|\Delta A\|^2\big], \quad \Delta A = A(X_ {l,c}+\Delta X^{(b)}_ {l,c})-A(X_ {l,c}),$$
> where $A(\cdot)$ is the mapping from the channel output to the final action, and $\Delta X^{(b)}_ {l,c}$ is the quantization noise.
>
> Assuming a local linearity for small perturbations, we apply a first-order Taylor expansion to approximate the action deviation:
> $$\Delta A \approx J_ {A,X_ {l,c}}\,\Delta X^{(b)}_ {l,c},$$
> where $J_ {A,X_ {l,c}}$ is the Jacobian of the action $A$ with respect to the channel output $X_ {l,c}$. Substituting this approximation into $S^{(b)}_ {l,c}$ yields:
> $$S^{(b)}_ {l,c} \approx \mathbb{E}\big[ \|J_ {A,X_ {l,c}}\Delta X^{(b)}_ {l,c}\|^2 \big] = \mathbb{E}\Big[ (\Delta X^{(b)}_ {l,c})^\top \underbrace{(J_ {A,X_ {l,c}}^\top J_ {A,X_ {l,c}})}_ {=:H_{l,c}} \Delta X^{(b)}_ {l,c} \Big].$$
>
> We then employ the common uniform quantization noise model, characterized by zero mean and an isotropic covariance structure:
> $$\mathbb{E}[\Delta X^{(b)}_ {l,c}] = 0,\quad \mathrm{Cov}(\Delta X^{(b)}_ {l,c}) = (\sigma^{(b)}_ {l,c})^2 I.$$
> Under this assumption, the expected quadratic form simplifies via the matrix trace property $\mathbb{E}[\mathbf{x}^\top H \mathbf{x}] = \mathrm{Tr}(H \mathrm{Cov}(\mathbf{x}))$:
> $$\mathbb{E}\big[ (\Delta X^{(b)}_ {l,c})^\top H_ {l,c}\,\Delta X^{(b)}_ {l,c} \big] = \mathrm{Tr}\!\big(H_ {l,c}\,\mathrm{Cov}(\Delta X^{(b)}_ {l,c})\big) = (\sigma^{(b)}_ {l,c})^2\mathrm{Tr}(H_ {l,c}).$$
> Since $\mathrm{Tr}(H_ {l,c})=\mathrm{Tr}(J_ {A,X_ {l,c}}^\top J_ {A,X_ {l,c}})=\|J_ {A,X_ {l,c}}\|_ F^2$ (the squared Frobenius norm of the Jacobian), we arrive at the final first-order approximation:
> $$S^{(b)}_ {l,c} \approx (\sigma^{(b)}_ {l,c})^2 \cdot \|J_ {A,X_ {l,c}}\|_ F^2.$$
> This theoretical derivation rigorously justifies our efficient proxy, which factors the sensitivity into a predictable quantization error term ($\sigma^{(b)}_ {l,c})^2$) and a task-dependent local sensitivity gain ($\|J_ {A,X_ {l,c}}\|_ F^2$).
>
> **We have also added related results and analysis in Appendix H in the revised manuscript.**

---

### Author Response · Authors · 2025-11-25
**General Response**

We sincerely thank the reviewers for their time and insightful feedback. We are greatly encouraged by the recognition of AutoQVLA’s practical utility and superior quantization effects in the VLA models. We also appreciate the constructive suggestions concerning architectural expansion, benchmark diversity, real-world validation, and refinement of methodological details.

To address these key concerns and rigorously demonstrate the superiority of our framework for VLA quantization, we conducted an extensive set of new experiments and in-depth analyses during the rebuttal period.
1.  **New Architectural Generalization :** We expanded validation to the UniVLA model, confirming AutoQVLA's sota performance across weight-only and weight-activation regimes. In the W4A16 setting, AutoQVLA preserved 99.9% of the original model's performance while significantly reducing VRAM usage. (Appendix D, Table 6).

2.  **Benchmark Expansion (CALVIN):** To demonstrate robustness against increased task complexity, we extended evaluation to the CALVIN benchmark. The compressed $\text{OpenVLA-OFT}$ ($\text{W}8\text{A}16$) model exhibited negligible performance degradation, successfully preserving stability required for complex long-horizon sequence planning tasks. (Appendix E, Table 7).

3.  **Real-World Validation on Physical Hardware ($\pi_{0}$):** We confirmed AutoQVLA's resilience to real-world variability using the $\pi_{0}$ baseline on real-world validation, encompassing single-arm and dual-arm coordination tasks. Even with sensor noise, the quantized policy preserved nearly all of the original performance, validating its practical robustness for deployment. (Appendix F, Table 8).

4.  **Methodological Superiority (Q-VLM) and Efficiency:** We compared against the VLM-specific Q-VLM baseline. AutoQVLA demonstrated superior acceleration ($1.42\times$ vs. $1.30\times$) and memory reduction, confirming our hardware advantage. We emphasize that AutoQVLA’s core is **Action-Centric Sensitivity**, which is intrinsically better suited than VLM-centric methods for mitigating the destructive accumulation of errors in VLA's closed-loop dynamic control.

5.  **Theoretical Analysis and Implementation Details:** We supplied the necessary methodological details, including the theoretical derivation for our first-order sensitivity proxy, providing a solid analytical foundation for our key design choices. We also detailed our custom CUDA kernel design and mixed-precision implementation, which provides the technical basis for achieving high speedup and performance retention in low-bit quantization.

6.  **Ablation Study on Calibration Robustness:** We included ablation studies on calibration set size, demonstrating that performance remains highly stable and robust to variations in the calibration distribution, proving the robustness of our estimation method. (Appendix G, Table 9).


These extensive additions collectively support our core claim that AutoQVLA is a robust, efficient, and methodologically superior quantization framework for Vision-Language-Action models, surpassing existing LLM and VLM baselines across generalization, efficiency, and task fidelity.

---

### Author Response · Authors · 2025-12-01
**Summary of Contributions and Rebuttal**

Dear PCs, SACS, ACs,

We would like to express our sincere appreciation for the reviewers’ dedication. Here we provide a general summary of the reviews and outline the efforts we have made during the discussion phase.

### **Claim of Contribution**

In this paper, we introduce **AutoQVLA**, the first *action-centric* quantization framework tailored to Vision-Language-Action (VLA) models. AutoQVLA performs fine-grained, channel-wise bit allocation driven directly by action-space sensitivity. It measures how quantizing each channel to different bit-widths perturbs the final continuous actions and estimates per-channel sensitivity, and then applies a global greedy demotion strategy that unifies quantization and pruning (0-bit) into a single optimization process. AutoQVLA achieves substantial VRAM reduction and speedup with minimal performance loss on 4 models(OpenVLA/OpenVLA-OFT/UniVLA/π0), 2 benchmarks(LIBERO/CALVIN), and real-world experiments.

### **Summary of Reviews and Responses**
We sincerely thank all reviewers for their detailed and insightful suggestions as well as the time invested during the rebuttal. These joint efforts have further improved AutoQVLA by enriching the content with additional experiments and clarifying the computational workflow. We are thrilled that **all four reviewers have unanimously expressed that this paper is worthy of acceptance.** The enthusiasm shown by Reviewer 9cSM is particularly encouraging as **they expressed a distinct intent to build upon our research by stating they can follow our excellent work right from the start.**

- **Reviewer kUNf** maintained a score of **6** and highlighted the novel quantization algorithm for VLA models. They also commended its empirical effectiveness.
- **Reviewer 9cSM** increased the rating from 4 **to 6** before the leak incident. This reviewer appreciated the fine-grained channel-wise sensitivity modeling and adaptive bit allocation.
- **Reviewer 2rUj** raised the score from 4 **to 6** before the leak incident. The main suggestion was to include real-world experiments and analysis. We responded by adding experiments on the $\pi_0$ model. The reviewer subsequently confirmed that these results addressed their concerns.
- **Reviewer RtD1** maintained a rating of **6**. They praised the well-motivated nature of our method. This reviewer also emphasized the significant reductions in memory requirements and inference latency.

We now address the specific points raised by the reviewers. All concerns have been carefully addressed in the table below.

| Reviewer | Reviewer's Concern/Questions | Author's Response |
| -- | -- | -- |
| Common   | Evaluations on more benchmarks, models and simulators.| We extended evaluation by adding UniVLA-7B, the CALVIN benchmark, and real-world experiments on a π0-based model. |
|    | Theoretical justification. | We added a rigorous theoretical derivation of our sensitivity metric. |
| kUNf     | Need more fine-grained ablations. | Detailed gate ratio ablation was already included in Appendix C . And we add an ablation on calibration set sizes  |
| 9cSM     | Need comparison with QVLM and Limited novelty relative to GPTQ. | We added comprehensive comparisons with Q-VLM on OpenVLA. We clarified the novelty in granularity and systematic quantification, action-centric optimization for VLA robustness and compatibility with pruning|
| 9cSM     | Surprisingly strong results under W4A4 settings. | We described our three-stage activation processing pipeline helps to improve the final performance. |
| 9cSM     | More details about method and result. | We provided implementation details of our custom low-bit CUDA kernels. We explained how we decompose mixed-precision W4A4 into homogeneous sub-operations. |
| 9cSM     | Task dependence of the action-space sensitivity metric. | We clarified that our sensitivity metric is intentionally task-dependent through the calibration distribution and that moving to a new task only requires a one-time offline recalibration. |
| 2rUj     | Robustness under real-world observation noise.| We added real-world experiments covering both simple and dexterous tasks and AutoQVLA maintains robust performance under real-world noise and complex manipulation conditions. |
| RtD1     | More details on the evaluation setup. Qualitative analysis of specific behaviors. | We clarified the exact evaluation setup behind Sec. 4.3/Fig. 3. We further added qualitative rollout analyses and examples in the revised manuscript. |



We are thrilled that all four reviewers have unanimously expressed that this paper is worthy of acceptance. Once again, we sincerely thank you for your support of our work and the generous time and effort you have devoted to this process!

---

### Meta-Review · Area_Chair_La4S · 2025-12-29

**Summary:**

This paper studies quantization of VLA models. While the initial submission had multiple weaknesses, especially around the extent of the experimental evaluation, this has been solidly addressed in the author response through experiments with additional base VLA models, in a new simulated domain, and on a real robot. I recommend acceptance.

**Reviewer Concerns:**

I think the main concerns were addressed.

**Reviewer Scores:**

I think the reviewers would have raised their score.

---

### Decision · Program_Chairs · 2026-01-26

Accept (Poster)